# Antibody levels following vaccination against SARS-CoV-2: associations with post-vaccination infection and risk factors in two UK longitudinal studies

Nathan J Cheetham[1]*, Milla Kibble[1,2,3], Andrew Wong[4], Richard J Silverwood[5], Anika Knuppel[4], Dylan M Williams[4,6], Olivia KL Hamilton[7], Paul H Lee[8], Charis Bridger Staatz[5], Giorgio Di Gessa[9], Jingmin Zhu[9], Srinivasa Vittal Katikireddi[7], George B Ploubidis[5], Ellen J Thompson[1,4], Ruth CE Bowyer[1,2,10], Xinyuan Zhang[1], Golboo Abbasian[1], Maria Paz Garcia[1], Deborah Hart[1], Jeffrey Seow[11], Carl Graham[11], Neophytos Kouphou[11], Sam Acors[11], Michael H Malim[11], Ruth E Mitchell[2], Kate Northstone[2], Daniel Major-Smith[2], Sarah Matthews[2], Thomas Breeze[2], Michael Crawford[2], Lynn Molloy[2], Alex SF Kwong[2,12], Katie Doores[11], Nishi Chaturvedi[4], Emma L Duncan[1,13], Nicholas J Timpson[2]*, Claire J Steves[1,13]*

[1]Department of Twin Research and Genetic Epidemiology, King's College London, London, United Kingdom; [2]Population Health Sciences, Bristol Medical School, University of Bristol, Bristol, United Kingdom; [3]Department of Applied Mathematics and Theoretical Physics, University of Cambridge, Cambridge, United Kingdom; [4]MRC Unit for Lifelong Health and Ageing, University College London, London, United Kingdom; [5]Centre for Longitudinal Studies, University College London, London, United Kingdom; [6]Department of Medical Epidemiology and Biostatistics, Karolinska Institutet, Stockholm, Sweden; [7]MRC/CSO Social and Public Health Sciences Unit, University of Glasgow, Glasgow, United Kingdom; [8]Department of Health Sciences, University of Leicester, Leicester, United Kingdom; [9]Department of Epidemiology and Public Health, University College London, London, United Kingdom; [10]AI for Science and Government, The Alan Turing Institute, London, United Kingdom; [11]Department of Infectious Diseases, King's College London, London, United Kingdom; [12]Division of Psychiatry, University of Edinburgh, Edinburgh, United Kingdom; [13]Guy's & St Thomas's NHS Foundation Trust, London, United Kingdom

*For correspondence:
nathan.cheetham@kcl.ac.uk
(NJC);
N.J.Timpson@bristol.ac.uk (NJT);
claire.j.steves@kcl.ac.uk (CJS)

## Abstract

**Background:** Severe acute respiratory syndrome coronavirus 2 (SARS-CoV-2) antibody levels can be used to assess humoral immune responses following SARS-CoV-2 infection or vaccination, and may predict risk of future infection. Higher levels of SARS-CoV-2 anti-Spike antibodies are known to be associated with increased protection against future SARS-CoV-2 infection. However, variation in antibody levels and risk factors for lower antibody levels following each round of SARS-CoV-2 vaccination have not been explored across a wide range of socio-demographic, SARS-CoV-2 infection and vaccination, and health factors within population-based cohorts.

**Methods:** Samples were collected from 9361 individuals from TwinsUK and ALSPAC UK population-based longitudinal studies and tested for SARS-CoV-2 antibodies. Cross-sectional sampling was undertaken jointly in April-May 2021 (TwinsUK, N=4256; ALSPAC, N=4622), and in TwinsUK only in

November 2021-January 2022 (N=3575). Variation in antibody levels after first, second, and third SARS-CoV-2 vaccination with health, socio-demographic, SARS-CoV-2 infection, and SARS-CoV-2 vaccination variables were analysed. Using multivariable logistic regression models, we tested associations between antibody levels following vaccination and: (1) SARS-CoV-2 infection following vaccination(s); (2) health, socio-demographic, SARS-CoV-2 infection, and SARS-CoV-2 vaccination variables.

**Results:** Within TwinsUK, single-vaccinated individuals with the lowest 20% of anti-Spike antibody levels at initial testing had threefold greater odds of SARS-CoV-2 infection over the next 6–9 months (OR = 2.9, 95% CI: 1.4, 6.0), compared to the top 20%. In TwinsUK and ALSPAC, individuals identified as at increased risk of COVID-19 complication through the UK 'Shielded Patient List' had consistently greater odds (two- to fourfold) of having antibody levels in the lowest 10%. Third vaccination increased absolute antibody levels for almost all individuals, and reduced relative disparities compared with earlier vaccinations.

**Conclusions:** These findings quantify the association between antibody level and risk of subsequent infection, and support a policy of triple vaccination for the generation of protective antibodies.

**Funding:** Antibody testing was funded by UK Health Security Agency. The National Core Studies program is funded by COVID-19 Longitudinal Health and Wellbeing – National Core Study (LHW-NCS) HMT/UKRI/MRC ([MC_PC_20030] and [MC_PC_20059]). Related funding was also provided by the NIHR 606 (CONVALESCENCE grant [COV-LT-0009]). TwinsUK is funded by the Wellcome Trust, Medical Research Council, Versus Arthritis, European Union Horizon 2020, Chronic Disease Research Foundation (CDRF), Zoe Ltd and the National Institute for Health Research (NIHR) Clinical Research Network (CRN) and Biomedical Research Centre based at Guy's and St Thomas' NHS Foundation Trust in partnership with King's College London. The UK Medical Research Council and Wellcome (Grant ref: [217065/Z/19/Z]) and the University of Bristol provide core support for ALSPAC.

## Editor's evaluation

This important study provides convincing evidence that a third SARS-CoV-2 vaccination elicits substantial increases in spike antibody responses, decreasing variability in titers observed after first and second doses, and also decreasing differences between groups at low and high-risk of low antibody responses. High antibody titers are subsequently associated with a lower incidence of infection. This paper is strong methodologically and will be of interest to clinicians and public health officials.

## Introduction

Immunological responses to severe acute respiratory syndrome coronavirus 2 (SARS-CoV-2) infection and SARS-CoV-2 vaccination vary between individuals and over time (*Shrotri et al., 2021a*; *Shrotri et al., 2021b*; *Wei et al., 2021b*). Within 2–4 weeks of infection, most individuals generate detectable levels of several antibody subtypes (immunoglobulin A, M, G) directed against different domains of the virus (Nucleocapsid protein, Spike protein, receptor-binding domain of Spike), which gradually decline over time (*Seow et al., 2020*; *Brochot et al., 2020*; *Wei et al., 2021a*; *Long et al., 2020*; *Gaebler et al., 2021*). Anti-Nucleocapsid antibodies are generated following infection but not by any current SARS-CoV-2 vaccines, while anti-Spike antibodies are generated following infection or vaccination. Levels of anti-Spike antibodies correlate with SARS-CoV-2-neutralising anti-receptor-binding domain antibody titre (*Perkmann et al., 2021*). A similar profile of antibody induction with subsequent waning is observed after vaccination against SARS-CoV-2 (*Shrotri et al., 2021a*; *Shrotri et al., 2021b*; *Matsunaga et al., 2022*; *Pegu et al., 2021*). Waning of antibody levels are likely correlated with observed reductions in vaccine effectiveness over time (*Pouwels et al., 2021*; *UKHSA, 2021*; *Lopez Bernal et al., 2021*). Reduced antibody neutralising activity and vaccine effectiveness have been observed for variants of concern in comparison to ancestral SARS-CoV-2 (*UKHSA, 2021*; *Collie et al., 2022*; *Yu et al., 2022*; *Nemet et al., 2022*; *Cheng et al., 2022*).

Anti-Spike antibody levels have been found to be inversely proportional to risk of future infection, and so identified as a correlate of protection (*Goldblatt et al., 2022a*; *Perry et al., 2022*; *Gilbert et al., 2022*; *Goldblatt et al., 2022b*; *Feng et al., 2021*; *Dimeglio et al., 2022a*; *Dimeglio et al.,*

**eLife digest** Vaccination against the virus that causes COVID-19 triggers the body to produce antibodies that help fight future infections. But some people generate more antibodies after vaccination than others. People with lower levels of antibodies are more likely to get COVID-19 in the future. Identifying people with low antibody levels after COVID-19 vaccination is important. It could help decide who receives priority for future vaccination.

Previous studies show that people with certain health conditions produce fewer antibodies after one or two doses of a COVID-19 vaccine. For example, people with weakened immune systems. Now that third booster doses are available, it is vital to determine if they increase antibody levels for those most at risk of severe COVID-19.

Cheetham et al. show that a third booster dose of a COVID-19 vaccine boosts antibodies to high levels in 90% of individuals, including those at increased risk. In the experiments, Cheetham et al. measured antibodies against the virus that causes COVID-19 in 9,361 individuals participating in two large long-term health studies in the United Kingdom. The experiments found that UK individuals advised to shield from the virus because they were at increased risk of complications had lower levels of antibodies after one or two vaccine doses than individuals without such risk factors. This difference was also seen after a third booster dose, but overall antibody levels had large increases. People who received the Oxford/AstraZeneca vaccine as their first dose also had lower antibody levels after one or two doses than those who received the Pfizer/BioNTech vaccine first. Positively, this difference in antibody levels was no longer seen after a third booster dose. Individuals with lower antibody levels after their first dose were also more likely to have a case of COVID-19 in the following months.

Antibody levels were high in most individuals after the third dose. The results may help governments and public health officials identify individuals who may need extra protection after the first two vaccine doses. They also support current policies promoting booster doses of the vaccine and may support prioritizing booster doses for those at the highest risk from COVID-19 in future vaccination campaigns.

*2022b*; *Khoury et al., 2021*). Goldblatt et al. estimated protective thresholds of 154 (95% CI: 42, 559) and 171 (95% CI: 57, 519) BAU/mL for wild-type and alpha variant SARS-CoV-2 respectively and an initial estimate range of 36–490 BAU/ml for delta variant (*Goldblatt et al., 2022b*), while *Feng et al., 2021* estimated 80% vaccine effectiveness against alpha variant for levels above 264 (95% CI: 108, 806) BAU/mL. Dimeglio et al. estimated much higher levels of more than 6000 BAU/mL were needed for protection against omicron variant BA.1, while no relationship was found between infection and antibody level for BA.2 (*Dimeglio et al., 2022b*).

Several clinical variables contribute to variation in antibody response following vaccination. Lower antibody levels following both first and second vaccinations have been observed in individuals with particular comorbidities (including cancer, renal disease, and hepatic disease; *Parry et al., 2021*; *Monin et al., 2021*; *Kearns et al., 2021*), individuals using immunosuppressant medications (*Shrotri et al., 2021a*; *Shrotri et al., 2021b*; *Parry et al., 2021*; *Monin et al., 2021*), and individuals identified from electronic health records data as of moderate or high risk of COVID-19 complications (according to UK government prior 'Shielded Patient List' criteria of conditions, ongoing treatments, and medications) (*Shrotri et al., 2021a*; *Shrotri et al., 2021b*; *GOV.UK, 2022b*). Studies testing for associations between antibody response and non-clinical factors such as socio-demographics have been more limited. Here, the use of longitudinal studies, with broader catalogues of bio-social data, are well suited to such investigations.

Here, we aimed to examine variables associated with variation in post-vaccination antibody levels, and the subsequent likelihood of post-vaccination infection. We measured the antibody levels of participants from two population-based longitudinal cohorts during the time of the UK vaccination roll-out: TwinsUK (in April-May 2021 and November 2021-January 2022) (*Verdi et al., 2019*) and Avon Longitudinal Study of Parents and Children (ALSPAC) (*Fraser et al., 2013*; *Boyd et al., 2013*) (April-May 2021 only). We aimed firstly to assess the relationship between anti-Spike antibody levels (identified as a correlate of protection against infection), measured after first or second vaccination in April-May 2021, and an outcome of subsequent post-vaccination infection over the following 6–9

months (identified through further serological evidence and/or self-reported COVID-19 from data collected in TwinsUK between November 2021 and January 2022). Secondly, we used peri-pandemic and historical data to investigate associations with an outcome of having relatively low antibody levels following first, second (ALSPAC and TwinsUK), or third (TwinsUK only) vaccination, for multiple socio-demographic, physical and mental health characteristics, prior SARS-CoV-2 infection, and vaccination history. Finally, we used twin-pair analysis within TwinsUK to probe genetic and environmental contributions to antibody level variation.

## Methods

### Study participants

TwinsUK is a UK-based national registry of monozygotic (MZ) and dizygotic (DZ) twins, with over 15,000 twins registered since 1992 (*Verdi et al., 2019*).

ALSPAC is a prospective population-based cohort of pregnant women with expected delivery dates between April 1991 and December 1992 who lived in Bristol, UK, and the nearby surrounding area; with follow-up of these women and their partners (collectively known as Generation 0, G0), and their children (Generation 1, G1), ever since (*Fraser et al., 2013*; *Boyd et al., 2013*). The initial cohort consisted of 14,541 pregnancies, with 13988 children alive at 1 year, and was later expanded when children were approximately age 7, to give a total of 15454 pregnancies, with 14,901 children alive at 1 year. Analyses herein were carried out solely with G0 participants due to low rates of vaccination among the G1 children generation at the time of initial serology.

During the COVID-19 pandemic, participants from both cohorts were invited to complete cohort-specific questionnaires and to submit blood samples via post for SARS-CoV-2 antibody testing. In the first round of coordinated testing in TwinsUK and ALSPAC, participants submitted samples in April and May 2021. This first testing round is referred to throughout as Q2 testing (from calendar year quarter 2 start date). Participants of TwinsUK were later invited for a second round of antibody testing with the same assay, with samples collected from November 2021 to January 2022. This round of antibody testing is referred to throughout as Q4 testing (from quarter 4 start date). Further details of COVID-19 questionnaires and antibody testing are given in following sections.

Inclusion and exclusion criteria were as follows. Individuals with unknown vaccination status at time of antibody testing were excluded from all analyses. For descriptive analysis of antibody levels versus time since vaccination, all individuals with known vaccination status were included. For analysis of variables associated with low antibody levels, individuals sampled fewer than 28 days since first vaccination, or fewer than 14 days since second or third vaccination, were excluded (these thresholds were chosen to allow sufficient time for an immunological response after each vaccine dose, based on previous studies *Shrotri et al., 2021a*; *Shrotri et al., 2021b*), while individuals with 77 days or more since first vaccination were excluded in case of misclassification due to unreported further vaccination (based on 11–12 week spacing between doses for majority of adults in the UK). In addition to the above criteria, for analysis of variables associated with post-vaccination infection within TwinsUK, individuals must have participated in Q2 antibody testing followed by either Q4 antibody testing and/or concurrent COVID-19 questionnaire.

### Questionnaires administered during the COVID-19 pandemic

TwinsUK COVID-19 questionnaires were administered in April-May 2020 (*Suthahar et al., 2021*), July-August 2020, October-December 2020, April-July 2021 (approximating first round of antibody testing, Q2), and November 2021-February 2022 (approximating second round of antibody testing, Q4). ALSPAC COVID-19 questionnaires were administered in April-May 2020 (*Northstone et al., 2020a*), May-July 2020 (*Northstone et al., 2020b*), October 2020 (*Northstone et al., 2021*), November 2020-March 2021 (approximating first round of antibody testing, Q2) (*Smith et al., 2021*), and July-December 2021.

Details of variables collected through cohort-specific pandemic questionnaires are provided in *Supplementary file 1*. Questions included self-reported SARS-CoV-2 infection and symptoms, results of SARS-CoV-2 testing, and vaccination status (date, dose number, manufacturer/brand) once the UK SARS-CoV-2 vaccination programme had commenced (8 December 2020). Questions made no distinction between pre-planned third vaccination for high-risk individuals and third vaccination given

as part of the wider community 'booster' campaign – as such we refer to third vaccination or triple-vaccinated individuals throughout. By virtue of the national vaccination roll-out policy (tiered by age and at-risk status), at Q2 participants may have received nought, one, or two vaccination doses; by Q4 some individuals had received a third dose.

As questionnaires were cohort-specific, assessed variables were not completely uniform (both question wording and collected data). Details for comparison are shown in *Supplementary file 1*.

## SARS-CoV-2 antibody testing

Q2 testing in TwinsUK and ALSPAC occurred along with an additional nine UK-based longitudinal studies who collected samples in unison as part of the UK National Core Studies Longitudinal Health & Wellbeing (NCS-LH&W) programme (*University College London, 2022*). Additional cohort-specific details and results for ALSPAC and Extended Cohort for E-health, Environment and DNA (EXCEED) are provided elsewhere (*Lee et al., 2021*; *Major-Smith et al., 2021*). Data availability in cohorts limited analysis to TwinsUK and ALSPAC.

For TwinsUK antibody testing in Q2 and Q4, invitation criteria were based on availability of email addresses and/or completion of previous COVID-19 questionnaires. ALSPAC invitation criteria are given in detail elsewhere (*Major-Smith et al., 2021*). For both cohorts, participants received finger-prick blood sample collection kits via post. Blood sample collection was self-administered. Samples were sent via post to either Pura Diagnostics or Eurofins County Pathology (partner laboratories of Thriva Ltd), who assayed samples and shared results with TwinsUK and ALSPAC. Quantitative IgG anti-Spike SARS-CoV-2 antibody levels and qualitative IgG anti-Nucleocapsid antibody status were assayed using CE-marked capillary blood Roche Elecsys Anti-SARS-CoV-2 immunoassays (*Roche, 2021*). Quantitative anti-Spike results were given in units per millilitre (U/mL), with a quantitative range of 0.4–250 U/mL for Q2 testing. For Q4 testing, samples were diluted to give an expanded quantitative range of 0.4–25000 U/mL, allowing quantitative discernment for higher levels at this timepoint. Tests had a positive threshold of 0.8 U/mL. One U/mL is equivalent to 1 unit of WHO standardised unit, binding antibody units per millilitre (BAU/mL) (WHO international standard: 20/136 *NIBSC, 2020*). Thus, we have quoted results in BAU/mL to aid comparison across studies. Anti-Nucleocapsid results were qualitative, with a positive result for a value greater than a cut-off unit = 1.

Additional antibody testing was also undertaken in-house for TwinsUK samples between April 2020 and April 2021. Quantitative enzyme-linked immunosorbent assay (ELISA) testing anti-Nucleocapsid and anti-Spike antibody levels were performed using previously published methods (*Seow et al., 2020*). These data were used to determine serology-based infection status prior to Q2 antibody testing.

## Identification of SARS-CoV-2 infection

### Assessment of prior SARS-CoV-2 infection, at time of antibody testing

Prior SARS-CoV-2 infection was classified with three distinct variables derived from self-reported questionnaire data or serological testing.

1. 'SARS-CoV-2 infection status (self-reported)': derived solely from self-reported COVID-19 infection and testing questionnaire data. The classification was primarily derived from responses to 'Do you think that you have or have had COVID-19?' in prior questionnaires. Classification options are given below:
   a. Confirmed case: 'Yes, confirmed by a positive test'.
   b. Suspected case: 'Yes, suspected by a doctor but not tested'.
   c. Suspected case: 'Yes, my own suspicions'.
   d. Unsure (TwinsUK only): 'Unsure'.
   e. No: 'No'. In TwinsUK questionnaires only, individuals were also asked to self-report any positive COVID-19 tests. Infection status of individuals who self-reported a positive test was classified as a confirmed case, irrespective of their answer to the question above.
2. 'SARS-CoV-2 infection status (serology-based)': derived from laboratory serological testing (Q2 [TwinsUK and ALSPAC], Q4 [TwinsUK only], and/or other within-cohort testing [TwinsUK only]), informed by self-reported vaccination status. We followed Centers for Disease Control and Prevention guidance on interpretation of anti-Nucleocapsid and anti-Spike results while accounting for vaccination status (*CDC, 2021*) as follows:

 a. Evidence of SARS-CoV-2 infection: A positive anti-Nucleocapsid result at any time or a positive anti-Spike result prior to vaccination.

 b. No evidence of SARS-CoV-2 infection: Negative anti-Nucleocapsid and anti-Spike result prior to vaccination, or negative anti-Nucleocapsid and positive anti-Spike result following vaccination (anti-Spike antibody assumed to be generated by vaccination).

3. 'Anti-Nucleocapsid antibody status': derived solely from laboratory serological testing (from Q2 or Q4 testing only). The classification was as follows:

 a. Positive: Positive anti-Nucleocapsid test result at Q2 or Q4 testing.

 b. Negative: Negative anti-Nucleocapsid test result at Q2 or Q4 testing.

From these variables, distinct measures of the proportion of individuals with evidence of prior SARS-CoV-2 infection, or 'natural infection', at time of Q2 and Q4 testing were quantified within both cohorts.

Thus, 'SARS-CoV-2 infection status (self-reported)' and 'SARS-CoV-2 infection status (serology-based)' variables identify individuals with any history of SARS-CoV-2 infection (who are not necessarily seropositive for anti-Nucleocapsid antibodies at time of testing), while 'Anti-Nucleocapsid antibody status' assesses the contemporaneous level of infection-induced antibody response.

## Assessment of post-vaccination SARS-CoV-2 infection

For analysis of variables associated with post-vaccination SARS-CoV-2 infection (performed within TwinsUK only), individuals with post-vaccination SARS-CoV-2 infections were identified using the following criteria:

1. A 'suspected case' or 'confirmed case' from 'SARS-CoV-2 infection status (self-reported)' variable at Q4 testing, with symptoms commencing after first vaccination. Infection and vaccination dates obtained from COVID-19 questionnaires.

2. A 'confirmed case' from 'SARS-CoV-2 infection status (self-reported)' variable at Q4 testing, with a self-reported positive antigen test dated after first vaccination. Infection and vaccination dates obtained from COVID-19 questionnaires.

3. A positive SARS-CoV-2 anti-Nucleocapsid result at Q4 testing after previous negative anti-Nucleocapsid results up to and including Q2, for individuals vaccinated at least once at Q2. The approximate date of infection is unknown for individuals who meet this criterion only.

Individuals meeting one or more of these criteria were considered as having post-vaccination infection. Individuals who did not meet any of these criteria were considered as controls (i.e., no post-vaccination infection). Individuals must have participated in TwinsUK Q4 antibody testing and/or concurrent COVID-19 questionnaire for post-vaccination infection to be determinable and for inclusion as controls or cases.

## Phenotypic data list

Variables from antibody testing and pandemic questionnaire data were supplemented with pre-pandemic socio-demographic and health variables for TwinsUK and ALSPAC analyses (details in *Supplementary file 1*). A full list of variables considered in analyses is given in *Table 1*.

## Statistical analyses

### Descriptive analysis of antibody levels after first, second, and third vaccination

Median, interquartile range, 10th and 5th percentile antibody levels were produced for univariate splits of adjustment and exposure variables listed in *Table 1*. Differences in median antibody levels (per Results) were tested using a two-sided Mann-Whitney U-test (*Mann and Whitney, 1947*). Trend in median antibody level versus number of weeks post-vaccination was tested using the Mann-Kendall trend test (*Mann, 1945*; *Kendall, 1975*).

### Association between SARS-CoV-2 infection and socio-demographic variables

Associations between SARS-CoV-2 infection, quantified from SARS-CoV-2 infection status (self-reported), SARS-CoV-2 infection status (serology-based), and anti-Nucleocapsid antibody status, and

**Table 1.** Phenotypic variables used in analyses.

Variables marked with an asterisk were outcome variables in logistic regression analyses; all other variables were adjustment or exposure variables. Variables only available in TwinsUK are notated as [TUK], and those only in ALSPAC as [ALSPAC].

| Variable group | Variable |
| --- | --- |
| Antibody levels | Anti-Spike level* |
| Socio-demographic | Age<br>Sex<br>Ethnicity<br>Local area deprivation (index of multiple deprivation, IMD [using national IMD rank decile/quintile]) (*GOV.UK, 2019*; *Northern Ireland Statistics and Research Agency, 2017*; *Gov.scot, 2020*; *GOV.WALES, 2021*)<br>Rural-urban classification [TUK] (*GOV.UK, 2022a*)<br>Highest educational attainment<br>Employment status |
| COVID-19 infection | SARS-CoV-2 infection status (self-reported)<br>SARS-CoV-2 infection status (serology-based)<br>Anti-Nucleocapsid antibody status<br>Post-vaccination SARS-CoV-2 infection [TUK]* |
| COVID-19 vaccination | Brand/manufacturer of first/second/third vaccination<br>Number of weeks between first/second/third vaccination and antibody sampling |
| Health indicators | Body mass index<br>Frailty index [TUK] (derived following *Searle et al., 2008*)<br>Frailty (PRISMA-7 assessment; *Raîche et al., 2008*) [ALSPAC]<br>Self-reported advised as on 'Shielded Patient List'<br>Self-rated health (5-point scale from 'poor' to 'excellent')<br>Prescribed immunosuppressant medication [TUK]<br>Self-reported immunocompromised [ALSPAC]<br>Anxiety (hospital anxiety and depression assessment scale (HADS) [TUK] *Zigmond and Snaith, 1983*, or 7-item generalised anxiety disorder scale (GAD-7) [ALSPAC] *Spitzer et al., 2006*, assessment)<br>Depression (HADS [TUK] or short mood and feelings questionnaire (SMFQ) [ALSPAC] *Turner et al., 2014* assessment)<br>Number of comorbidities from: anxiety/depression, diabetes, cancer, hypertension, heart disease |
| Individual comorbidities | Anxiety<br>Arthritis (any) [TUK]<br>Asthma<br>Atrial fibrillation [TUK]<br>Cancer (any)<br>Depression<br>Diabetes (any)<br>Heart disease<br>High cholesterol [TUK]<br>Hypertension<br>Lung disease<br>Osteoporosis [TUK]<br>Rheumatoid arthritis [TUK]<br>Stroke [TUK] |
| Comorbidity domains | Cardiac disease [TUK]<br>Cardiac risk factors [TUK]<br>Neurological disease<br>Subjective memory impairment [TUK] |

age, sex, ethnicity, local area deprivation, and rural-urban classification were tested using the chi-square test of independence.

## Logistic regression analyses

Within TwinsUK only, univariable and multivariable logistic regression were used to test associations between an outcome of post-vaccination SARS-CoV-2 infection and exposure variables related to: Q2 anti-Spike antibody levels; socio-demographics; COVID-19 infection; COVID-19 vaccination. In TwinsUK and ALSPAC, multivariable logistic regression was also performed to test associations between the outcome of low anti-Spike antibody levels (as defined below) after each round of vaccination (after first and second vaccinations for both TwinsUK and ALSPAC, and after third vaccination for TwinsUK only) and all exposure variables previously listed.

Each model included the outcome variable, a single exposure variable of interest, and a set of adjustment variables. Individual exposure variables of interest were tested in sequence, fitting a separate logistic regression model for each combination of outcome, adjustment, and exposure variables. Only individuals with complete data for the given model were included. For each categorical variable within logistic regression models, reference categories were chosen based on the normative, modal, maximum or minimum value/category, as appropriate (reference categories given in *Supplementary file 1*). Within TwinsUK models only, the HC3 estimator of logistic regression coefficient standard errors was used to account for heteroskedasticity (which biases conventional standard errors in analysis of related twin-pairs; *Hayes and Cai, 2007*; *statsmodels, 2021*; *Farbmacher and Kögel, 2017*). (Two-sided) p-values were corrected for multiple testing using the Benjamini/Hochberg p-value adjustment (*Benjamini and Hochberg, 1995*).

An outcome of post-vaccination SARS-CoV-2 infection was identified using the criteria previously described. An a priori outcome of 'low anti-Spike antibody levels' was defined relatively within each group stratified by vaccination status (single-, double-, triple-vaccinated within TwinsUK, and single-, double-vaccinated within ALSPAC) and assigned to individuals in the lowest 10% of anti-Spike antibody levels. As such the anti-Spike threshold value used to define low levels varied between models. Most double-vaccinated individuals at Q2 testing had antibody levels above the upper assay limit of 250 BAU/mL (TwinsUK: 92%, ALSPAC: 92%). Thus, a threshold of <250 BAU/mL was used instead of 10% to identify low antibody levels after second vaccination at Q2 testing, corresponding to the lowest 8% in both TwinsUK and ALSPAC. In total, for each exposure variable, there were four TwinsUK models and two ALSPAC models.

Multivariable models testing association between post-vaccination SARS-CoV-2 infection and anti-Spike antibody levels used the following sets of adjustment variables: (1) number of weeks since most recent vaccination; (2) age, sex, number of weeks since most recent vaccination. Multivariable models testing association between post-vaccination SARS-CoV-2 infection and socio-demographic variables used the following sets of adjustment variables: (1) age; (2) age, SARS-CoV-2 infection status (serology-based); (3) age, sex, SARS-CoV-2 infection status (serology-based). Multivariable models testing associations with low anti-Spike antibody levels used the following set of adjustment variables: age, sex, most recent vaccine received, and number of weeks since most recent vaccination. Adjustment variables were chosen based on relatively large effects observed in preliminary descriptive analysis.

## Twin-pair analyses

To assess the relationship between zygosity and relatedness on variation in antibody levels between pairs of individuals after third vaccination within TwinsUK, antibody level differences were calculated for all pairs of MZ and DZ twins, and within all combinations of non-related pairs. The difference between the resulting median pair differences within MZ, DZ, and non-related pairs were tested using the two-sided Mann-Whitney U-test.

For variables associated with low antibody levels (from logistic regression analyses), within-twin-pair associations with unadjusted anti-Spike antibody levels after third vaccination were tested using 'within-between' generalised linear mixed effects models. Such models implicitly control for pair-specific shared genetic and environmental factors by design and are commonly used in twin-pair studies as described elsewhere (*Carlin et al., 2005*). The pseudonymised family identifier variable was fitted as a random effect, allowing intercept to vary for each twin-pair. For the exposure variable

of interest, twin-pair mean values and difference-to-twin-pair-mean values were calculated and both included as 'between-pair' and 'within-pair' variables in models, respectively. Age, sex, number of weeks since third vaccination, brand of vaccine received for third vaccination, and SARS-CoV-2 infection status (serology-based) were also included in models as adjustment variables. For each exposure variable, separate models were fitted for MZ and DZ twin-pairs. Differences between 'between-pair' and 'within-pair' coefficients were tested using a Wald test. Unpaired single twins and individuals without data for all variables were excluded from the given model.

## Software

TwinsUK analyses were performed using python v3.8.8 (*Van Rossum and Drake, 2009*) and packages: numpy v1.20.1, pandas v1.2.4, statsmodels v0.12.2, scipy v1.6.2, scikit-learn v0.24.1, matplotlib v3.3.4, pymannkendall v1.4.2, seaborn v0.11.1. ALSPAC analyses were performed using python v3.9.7 and packages: numpy v1.20.3, pandas v1.3.4, matplotlib v3.4.3, and seaborn v0.11.2, and R v4.1.2 (*R Development Core Team, 2021*) and packages: plyr v1.8.6, dplyr v1.0.7, and broom v0.7.11.

## Results

### Cohort characteristics

Antibody levels were measured in 9361 individuals at two timepoints – 4256 individuals from TwinsUK and 4622 individuals from ALSPAC during April and May 2021 (referred to throughout as Q2 [calendar year quarter 2] testing), and 3575 individuals from TwinsUK in follow-up testing from November 2021 to January 2022 (referred to throughout as Q4 [quarter 4] testing). Response rates, as the percentage who returned sample after consenting and being sent a sample collection kit, were as follows: TwinsUK Q2: 87%, TwinsUK Q4: 80%, ALSPAC Q2: 79%. Flow charts showing identification of analysis samples are given in *Figure 1—figure supplements 1–3*. Results of antibody testing and selected characteristics are summarised in *Table 2* (with extended characteristics given in *Supplementary file 2*). Consistent with the tiered UK vaccination campaign, individuals who had received more vaccinations at either timepoint were older, more likely to be on the UK 'Shielded Patient List' (*GOV.UK, 2022b*), and had lower self-rated health, compared with those with fewer vaccinations. Participants were predominantly female and the vast majority were of white ethnicity in both cohorts, consistent with the broader composition of both cohorts. Prevalence of SARS-CoV-2 infection differed according to the measure of infection, either from self-report or from serological testing, and varied by vaccination status, socio-demographic variables, and between the two timepoints examined (*Table 2*, *Supplementary file 3*, *Figure 1—figure supplement 4*).

### Antibody levels after first, second, and third vaccination

Considering firstly data from Q4 testing undertaken within TwinsUK only, cross-sectional antibody levels following third vaccination were much greater and more sustained, with less inter-individual variability, compared to levels for those with fewer vaccinations. The median anti-Spike antibody levels in individuals who had received a third vaccination (unadjusted for time since vaccination) were over 10-fold higher than for individuals after second vaccination: 13700 BAU/mL after third vaccination, 1300 BAU/mL after second vaccination, 50 BAU/mL after first vaccination (*Figure 1*, detailed univariable splits of anti-Spike levels given in *Supplementary file 4*). There were also large increases in absolute levels for individuals at the bottom of the antibody level distribution after third vaccination, with 90% having level greater than 5000 BAU/mL, close to the estimated 6000 BAU/mL threshold estimated to confer partial protection against the omicron variant (*Dimeglio et al., 2022b*). The antibody level distribution after third vaccination was relatively narrower compared with earlier vaccination (median:IQR ratios of 0.54, 0.27, and 0.89 among Q2 single-vaccinated, Q4 double-vaccinated, and Q4 triple-vaccinated sub-samples, respectively), with smaller scale-factor differences between those with median and lowest levels (median:10th percentile ratios of 5.6, 11.8, and 2.7 among Q2 single-vaccinated, Q4 double-vaccinated, and Q4 triple-vaccinated sub-samples, respectively).

Considering antibody levels versus time since vaccination: within TwinsUK Q4 results, median antibody levels up to 16 weeks since third vaccination were highest in individuals sampled 2–3 weeks after vaccination (median: 24600 BAU/mL, n=203) (*Figure 2*). Although median antibody levels decreased

**Table 2.** Sample characteristics.

Antibody level values and characteristics for TwinsUK and Avon Longitudinal Study of Parents and Children (ALSPAC) individuals sampled in Q2 and Q4 antibody collections. Individuals are stratified by vaccination status at time of sampling. Data shown for individuals sampled at least 4 weeks after first vaccination, and at least 2 weeks after second or third vaccination to allow time for antibody generation. The anti-Spike antibody level assay range is 0.4–250 BAU/mL for Q2 results and 0.4–25,000 BAU/mL for Q4 results, with a positive threshold of 0.8 BAU/mL. Categories with fewer than five individuals are suppressed.

| Cohort | TwinsUK | | | | | ALSPAC | | | | | |
|---|---|---|---|---|---|---|---|---|---|---|---|
| Testing period | Q2 | Q4 | Q2 | Q2 | Q2 | Q4 | Q4 | Q2 | Q2 | Q2 | Q2 |
| Vaccination status | All results | All results | Not vaccinated | Single-vaccinated | Double-vaccinated | Double-vaccinated | Triple-vaccinated | All results | Not vaccinated | Single-vaccinated | Double-vaccinated |
| n | 4256 | 3575 | 330 | 1375 | 748 | 691 | 1937 | 1779 | 36 | 1459 | 284 |
| Age (years): Median (IQR) | 63.0 (49.0, 72.0) | 63.0 (51.0, 72.0) | 38.0 (31.0, 44.0) | 63.0 (56.0, 69.0) | 70.0 (56.0, 77.0) | 49.0 (38.0, 59.0) | 69.0 (60.0, 74.0) | 60.0 (57.0, 62.0) | 57.5 (52.75, 62.25) | 60.0 (57.0, 63.0) | 59.0 (56.0, 61.0) |
| Sex: Male, n (%) | 518/4255 (12.2%) | 447/3574 (12.5%) | 48/330 (14.5%) | 178/1375 (12.9%) | 88/748 (11.8%) | 103/691 (14.9%) | 225/1937 (11.6%) | 451/1779 (25.4%) | 8/36 (22.2%) | 397/1459 (27.2%) | 46/284 (16.2%) |
| Ethnicity: Other than White, n (%) | 118/4219 (2.8%) | 96/3536 (2.7%) | 16/329 (4.9%) | 29/1368 (2.1%) | 19/739 (2.6%) | 26/686 (3.8%) | 39/1914 (2.0%) | 26/1779 (1.5%) | <5 | 20/1459 (1.4%) | 5/284 (1.8%) |
| BMI: Median (IQR) | 24.75 (22.15, 27.99) | 24.76 (22.2, 27.98) | 22.72 (20.94, 25.42) | 24.86 (22.27, 28.28) | 24.99 (22.23, 28.07) | 23.91 (21.62, 27.58) | 24.87 (22.3, 27.87) | 25.7 (23.23, 28.7) | 25.63 (23.13, 28.04) | 25.71 (23.25, 28.77) | 25.65 (23.02, 28.6) |
| Advised on 'Shielded Patient List': Yes, n (%) | 341/4109 (8.3%) | 279/3530 (7.9%) | 8/329 (2.4%) | 82/1374 (6.0%) | 86/748 (11.5%) | 23/691 (3.3%) | 190/1936 (9.8%) | 67/1754 (3.8%) | <5 | 45/1443 (3.1%) | 22/276 (8.0%) |
| Self-rated health: Poor, Fair, n (%) | 357/4082 (8.7%) | 290/3407 (8.5%) | 15/316 (4.7%) | 134/1364 (9.8%) | 60/737 (8.1%) | 42/656 (6.4%) | 168/1871 (9.0%) | 167/1778 (9.4%) | <5 | 137/1459 (9.4%) | 28/283 (9.9%) |
| Zygosity: Monozygotic, n (%) | 2722/4253 (64.0%) | 2280/3573 (63.8%) | 248/328 (75.6%) | 883/1375 (64.2%) | 459/748 (61.4%) | 490/689 (71.1%) | 1170/1937 (60.4%) | – | – | – | – |
| Anti-Spike antibody level value (BAU/mL): Median (IQR) | 80.78 (18.55, 250.0) | 10403.0 (3510.0, 20224.0) | 0.4 (0.4, 0.4) | 53.3 (22.72, 121.2) | 250.0 (250.0, 250.0) | 1317.0 (337.0, 5202.5) | 13694.0 (8153.0, 23543.0) | 58.93 (21.25, 247.5) | 10.53 (0.4, 48.69) | 43.42 (17.98, 106.65) | 250.0 (250.0, 250.0) |
| Anti-Spike antibody status: Positive, n (%) | 3372/3912 (86.2%) | 3423/3445 (99.4%) | 79/330 (23.9%) | 1357/1375 (98.7%) | 745/748 (99.6%) | 690/691 (99.9%) | 1936/1937 (99.9%) | 1745/1779 (98.1%) | 23/36 (63.9%) | 1440/1459 (98.7%) | 282/284 (99.3%) |
| Anti-Nucleocapsid antibody status, Q2: Positive, n (%) | 460/3893 (11.8%) | 333/2887 (11.5%) | 60/329 (18.2%) | 156/1368 (11.4%) | 87/743 (11.7%) | 85/565 (15.0%) | 160/1624 (9.9%) | 167/1757 (9.5%) | <5 | 133/1438 (9.2%) | 31/283 (11.0%) |
| Anti-Nucleocapsid antibody status, Q4: Positive, n (%) | 524/2998 (17.5%) | 618/3447 (17.9%) | 80/290 (27.6%) | 197/1130 (17.4%) | 95/602 (15.8%) | 179/691 (25.9%) | 263/1937 (13.6%) | – | – | – | – |

*Table 2 continued on next page*

*Table 2 continued*

| Cohort | TwinsUK | | | | | | ALSPAC | | | |
|---|---|---|---|---|---|---|---|---|---|---|
| **Testing period** | Q2 | Q4 | | | | | Q2 | | | |
| **Vaccination status** | All results | All results | Not vaccinated | Single-vaccinated | Double-vaccinated | Triple-vaccinated | All results | Not vaccinated | Single-vaccinated | Double-vaccinated |
| Weeks since first vaccination: Median (IQR) | 10.0 (6.0, 12.0) | 42.0 (38.0, 45.0) | −5.0 (−8.0, −3.0) | 8.0 (6.0, 9.0) | – | – | – | – | 6.0 (5.0, 8.0) | – |
| First vaccination received: AZD1222 (Oxford/AZ), n (%) | 2124/3591 (59.1%) | 1980/3378 (58.6%) | 70/275 (25.5%) | 1103/1374 (80.3%) | – | – | – | – | 1235/1459 (84.6%) | – |
| First vaccination received: BNT162b2 (Pfizer BioNTech), n (%) | 1410/3591 (39.3%) | 1336/3378 (39.6%) | 170/275 (61.8%) | 266/1374 (19.4%) | – | – | – | – | 224/1459 (15.4%) | – |
| Weeks since second vaccination: Median (IQR) | −1.0 (−4.0, 2.0) | 32.0 (28.0, 34.0) | – | – | 3.0 (2.0, 5.0) | 33.0 (31.0, 35.0) | – | – | – | 4.0 (2.0, 6.0) |
| Second vaccination received: AZD1222 (Oxford/AZ), n (%) | 1858/3266 (56.9%) | 1888/3275 (57.6%) | – | – | 212/748 (28.3%) | 1065/1934 (55.1%) | – | – | – | 50/284 (17.6%) |
| Second vaccination received: BNT162b2 (Pfizer BioNTech), n (%) | 1357/3266 (41.5%) | 1330/3275 (40.6%) | – | – | 532/748 (71.1%) | 858/1934 (44.4%) | – | – | – | 234/284 (82.4%) |
| Weeks since third vaccination: Median (IQR) | −28.0 (−30.0, −26.0) | 5.0 (3.0, 7.0) | – | – | – | 5.0 (4.0, 8.0) | – | – | – | – |
| Third vaccination received: mRNA-1273 (Moderna), n (%) | 293/2149 (13.6%) | 337/2400 (14.0%) | – | – | – | 203/1903 (10.7%) | – | – | – | – |
| Third vaccination received: BNT162b2 (Pfizer BioNTech), n (%) | 1828/2149 (85.1%) | 2026/2400 (84.4%) | – | – | – | 1677/1903 (88.1%) | – | – | – | – |
| SARS-CoV-2 infection status (serology-based) at time of antibody testing: Evidence of natural infection, n (%) | 891/4190 (21.3%) | 977/3561 (27.4%) | 98/330 (29.7%) | 304/1375 (22.1%) | 157/748 (21.0%) | 464/1937 (24.0%) | 187/1757 (10.6%) | 23/36 (63.9%) | 133/1438 (9.2%) | 31/283 (11.0%) |
| SARS-CoV-2 infection status (self-reported), Q2: Suspected case, n (%) | 477/4092 (11.7%) | 399/3428 (11.6%) | 35/320 (10.9%) | 183/1365 (13.4%) | 67/739 (9.1%) | 197/1882 (10.5%) | 302/1675 (18.0%) | 5/33 (15.2%) | 240/1374 (17.5%) | 57/268 (21.3%) |

*Table 2 continued on next page*

Table 2 continued

| Cohort | TwinsUK | | | | | | | ALSPAC | | | |
|---|---|---|---|---|---|---|---|---|---|---|---|
| **Testing period** | Q2 | Q4 | Q2 | | | Q4 | | Q2 | | | |
| **Vaccination status** | All results | All results | Not vaccinated | Single-vaccinated | Double-vaccinated | Double-vaccinated | Triple-vaccinated | All results | Not vaccinated | Single-vaccinated | Double-vaccinated |
| SARS-CoV-2 infection status (self-reported), Q2: Confirmed case, n (%) | 597/4092 (14.6%) | 492/3428 (14.4%) | 57/320 (17.8%) | 218/1365 (16.0%) | 112/739 (15.2%) | 107/662 (16.2%) | 256/1882 (13.6%) | 40/1675 (2.4%) | <5 | 29/1374 (2.1%) | 11/268 (4.1%) |
| SARS-CoV-2 infection status (self-reported), Q4: Suspected case, n (%) | 478/4134 (11.6%) | 404/3543 (11.4%) | 34/330 (10.3%) | 183/1375 (13.3%) | 70/748 (9.4%) | 78/691 (11.3%) | 204/1936 (10.5%) | – | – | – | – |
| SARS-CoV-2 infection status (self-reported), Q4: Confirmed case, n (%) | 817/4134 (19.8%) | 751/3543 (21.2%) | 92/330 (27.9%) | 306/1375 (22.3%) | 145/748 (19.4%) | 202/691 (29.2%) | 357/1936 (18.4%) | – | – | – | – |

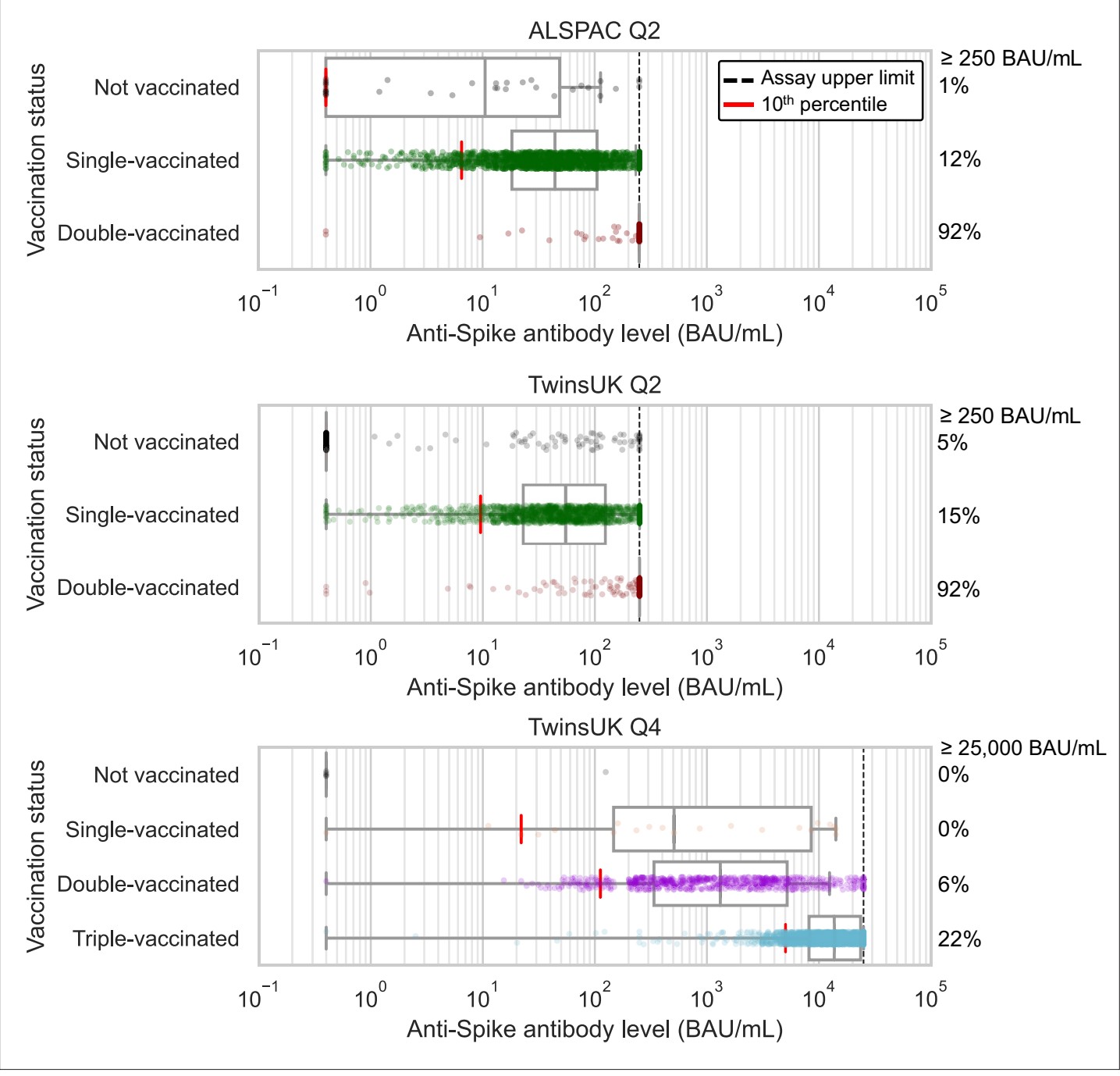

**Figure 1.** Anti-Spike antibody levels stratified by cohort and vaccination status at Q2 and Q4 antibody testing. Dot and box plots showing distribution of anti-Spike antibody levels within Avon Longitudinal Study of Parents and Children (ALSPAC) and TwinsUK, for those not vaccinated or individuals single-, double- or triple-vaccinated at time of sampling. Data shown for individuals sampled at least 4 weeks after first vaccination, and at least 2 weeks after second or third vaccination to allow time for antibody generation. Length of box plot whiskers are limited to 1.5 times the interquartile range. Red lines show 10th percentile levels. Assay upper limit is shown by black dotted lines, with 0.4–250 BAU/mL range for Q2 results and 0.4–25000 BAU/mL for Q4 results, with a positive threshold of 0.8 BAU/mL. Percentage of values above assay upper limit is given on right side of plots.

The online version of this article includes the following figure supplement(s) for figure 1:

**Figure supplement 1.** Flow chart showing identification of analysis samples from Q2 antibody testing within TwinsUK.

**Figure supplement 2.** Flow chart showing identification of analysis samples from Q4 antibody testing within TwinsUK.

**Figure supplement 3.** Flow chart showing identification of analysis samples from Q2 antibody testing within Avon Longitudinal Study of Parents and Children (ALSPAC).

*Figure 1 continued on next page*

*Figure 1 continued*

**Figure supplement 4.** SARS-CoV-2 infection prevalence by socio-demographic factors in TwinsUK.

between 2 and 8 weeks after third vaccination, there was no evidence of further decline between 8 and 16 weeks (Mann-Kendall trend test in median levels at 8+ weeks, p=0.60), and high absolute levels of antibodies were sustained (8+ weeks median = 9200 BAU/mL [IQR: 5800–16000 BAU/mL], n=519). These cross-sectional trends in median antibody levels versus time since third vaccination persisted when stratifying by age and other variables. Similarly, for individuals sampled 13–33 weeks after second vaccination, longer time since vaccination was also associated with lower antibody levels.

From Q2 results, antibody levels peaked at 9 weeks after first vaccination in both TwinsUK and ALSPAC. After second vaccination, median levels breached the 250 BAU/mL assay limit from 2 weeks onwards, precluding further time assessment.

## Factors associated with recorded post-vaccination infection in TwinsUK

Given the large variability in antibody response after first vaccination (*Figure 1*), we investigated whether a lower antibody response after first vaccination associated with post-vaccination 'breakthrough' infection, as evidenced by self-report (suspected or confirmed case) and/or serological testing (positive anti-Nucleocapsid test after vaccination). Within TwinsUK, post-vaccination SARS CoV-2 infection (between first vaccination and Q4 testing) was recorded in 276 of 2993 (9.2%) individuals (further details related to post-vaccination infection given in *Supplementary file 5*). Among those tested at Q2 while single-vaccinated, individuals with lower antibody levels had increased risk of subsequent infection over the next 6–9 months (*Table 3*). After controlling for age, sex, and number of weeks since vaccination, those with anti-Spike levels in the lowest 80% within the sample, <164 BAU/mL, had two- to threefold odds of post-vaccination infection than those in the highest quintile, ≥164 BAU/mL, with those in the lowest quintile, <18 BAU/mL, having the largest effect size (OR = 2.9 [95% CI: 1.4–6.0], p=0.02). Odds of post-vaccination infection was also found to be lower in older age groups (e.g., 80+ versus 18–49, OR = 0.18 [95% CI: 0.07–0.44], p=0.002), those with serological evidence of SARS-CoV-2 infection prior to Q2 testing versus those without (OR = 0.46 [95% CI: 0.32–0.67], p=0.0009), and for those who were retired versus employed (OR = 0.49 [95% CI: 0.33–0.74], p=0.01) (full multivariable results in *Supplementary file 6*).

## Factors associated with lower antibody levels within TwinsUK and ALSPAC

We tested for associations with having lower antibody levels after each round of vaccination. Lower antibody levels were defined as the lowest 10% within each sub-sample of cohort, testing round and vaccination status (<250 BAU/mL threshold corresponding to lowest 8% in both TwinsUK and ALSPAC used for Q2 double-vaccinated sub-samples where assay limit did not allow lowest 10% to be identified). Relative thresholds were used rather than absolute values due to the variation in reported thresholds between studies and for different SARS-CoV-2 variants, while the more general principal that antibody levels are inversely correlated with risk of infection has remained consistent throughout the COVID-19 pandemic. Increased odds of lower antibody levels were consistently observed across multiple vaccination rounds in TwinsUK and/or ALSPAC (*Figure 3*) for the following health-related variables:

1. Those advised as being on the UK 'Shielded Patient List' (*NHS Digital, 2021*; *NHS Digital, 2022*). For example, for lowest 10% after first vaccination, TwinsUK: (OR = 4.0, [95% CI: 2.2–7.4], p=0.0001), ALSPAC: (OR = 4.1, [95% CI: 1.8–9.5], p=0.02).
2. Those with poorer self-rated health. For example, for lowest 10% after first vaccination in TwinsUK: (OR = 1.4, [95% CI: 1.1–1.6], p=0.02), for a –1 step on an ordinal 1–5 (poor-excellent) scale.
3. Those with indicators of immunosuppression. For example, for lowest 10% after second vaccination in TwinsUK: (OR = 4.2, [95% CI: 1.9–9.5], p=0.006), or for lowest 10% after first vaccination in ALSPAC: (OR = 6.2, [95% CI: 2.7–14.5], p=0.001).

Results for all exposure variables are presented *Supplementary file 7*.

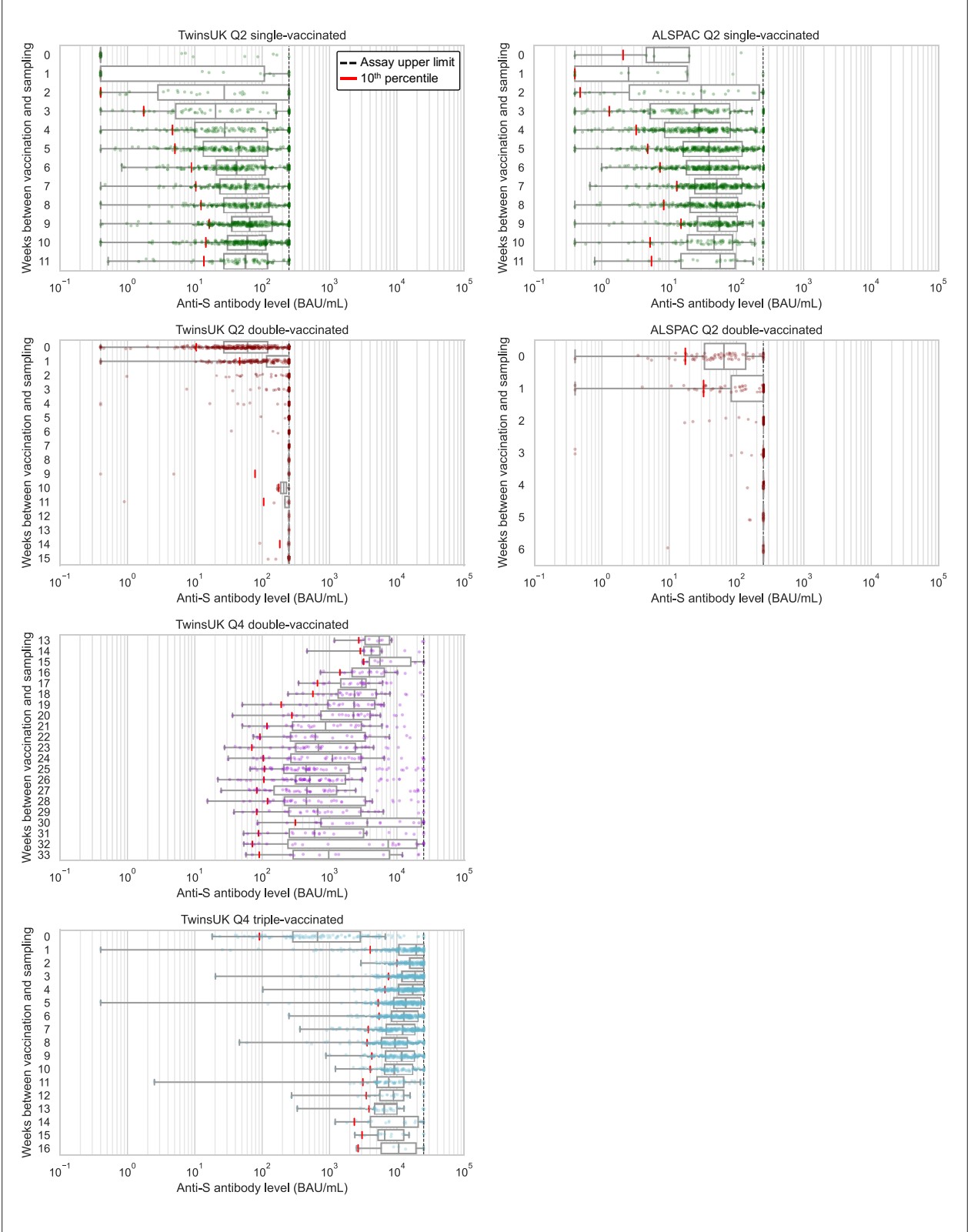

**Figure 2.** Anti-Spike antibody levels versus time since most recent vaccination, stratified by cohort and vaccination status at Q2 and Q4 antibody testing. Dot and box plots showing distribution of anti-Spike (anti-S) antibody levels within unvaccinated, single-, double- and triple-vaccinated individuals within Avon Longitudinal Study of Parents and Children (ALSPAC) (Q2 testing) and TwinsUK (Q2 and Q4 testing), plotted against the number of weeks since most recent vaccination at time of sampling. Length of box plot whiskers are limited to 1.5 times the interquartile range. Red lines show

*Figure 2 continued on next page*

*Figure 2 continued*

10th percentile levels. Assay upper limit is shown by black dotted lines, with 0.4–250 BAU/mL range for Q2 results and 0.4–25000 BAU/mL for Q4 results, with a positive threshold of 0.8 BAU/mL. X-axes are limited to weeks with results for five or more individuals, noting TwinsUK Q4 second vaccination sub-plot begins at 13 weeks since vaccination.

Individuals in both cohorts who received the AZD1222 (Oxford/AstraZeneca) vaccine versus BNT162b2 (Pfizer BioNTech) were more likely to have lower antibody levels after first vaccination (for lowest 10% in TwinsUK: (OR = 3.1, [95% CI: 1.5–6.4], p=0.02), and ALSPAC: (OR = 3.2, [95% CI: 1.4–7.7], p=0.09)), and second vaccination (for lowest 8% in TwinsUK Q2: (OR = 3.0, [95% CI: 1.4–6.2], p=0.03), TwinsUK Q4: (OR = 45.7, [95% CI: 5.6–372], p=0.001), and ALSPAC: (OR = 20.3, [95% CI: 6.4–64.7], p=0.0001)). However, receiving AZD1222 at second vaccination was not associated with lower antibody levels after third vaccination in TwinsUK (for lowest 10%, (OR = 1.1, [95% CI: 0.8–1.6], p=0.8)). Those with longer time since vaccination at time of sampling had increased odds of lower antibody levels after second and third vaccination, while individuals sampled later after first vaccination had decreased odds of lower antibody levels. Lower likelihood of lower antibody levels was seen across multiple rounds of vaccination within TwinsUK for those with evidence of SARS-CoV-2 infection prior to antibody testing, either through serological testing (e.g., outcome: lowest 10% after third vaccination (OR = 0.45, [95% CI: 0.28–0.71], p=0.004)) or self-reported confirmed cases (e.g., outcome: lowest 10% after third vaccination (OR = 0.25, [95% CI: 0.13–0.45], p=0.0001)), but not for self-reported suspected cases.

Less consistent associations (i.e., not observed across more than one round of vaccination) with increased likelihood of lower antibody levels were seen in TwinsUK for several other variables: very frail, high multimorbidity (three or more of five selected comorbidities), rheumatoid arthritis, employment status of permanently or long-term sick or disabled, and lower educational attainment (*Supplementary file 7*). No clear associations with lower antibody levels were seen with age, sex, or BMI in either TwinsUK or ALSPAC.

## Twin-pair analysis in TwinsUK after third vaccination

Within TwinsUK, pairs of identical MZ twins showed smaller average intra-pair anti-Spike antibody level differences after third vaccination versus non-identical DZ twin-pairs (median twin-pair difference

**Table 3.** Association between post-vaccination infection and anti-Spike antibody levels within TwinsUK.

Logistic regression model results, testing association between post-vaccination infection, and Q2 anti-Spike antibody levels in single-vaccinated individuals within TwinsUK. Reference category was a Q2 antibody level in quintile 5 (highest 20%). Results present odds ratios, unadjusted 95% confidence intervals, and p-values adjusted for multiple testing.

| Q2 antibody level | Post-vaccination infection incidence rate (%) | Unadjusted OR (95% CI), p-value | Adjusted for: Weeks since vaccination OR (95% CI), p-value | Adjusted for: Age, sex, weeks since vaccination OR (95% CI), p-value |
|---|---|---|---|---|
| Quintile 1 (lowest 20%): 0.4–18 BAU/mL | 32/233 (13.7%) | 3.23 (1.58–6.58), p=0.009 | 2.85 (1.39–5.86), p=0.03 | 2.93 (1.42–6.04), p=0.02 |
| Quintile 2: 18–40 BAU/mL | 20/226 (8.8%) | 1.97 (0.92–4.21), p=0.11 | 2.04 (0.94–4.43), p=0.08 | 2.15 (0.99–4.68), p=0.06 |
| Quintile 3: 40–73 BAU/mL | 21/239 (8.8%) | 1.95 (0.92–4.15), p=0.11 | 2.26 (1.04–4.92), p=0.06 | 2.41 (1.11–5.27), p=0.04 |
| Quintile 4: 73–164 BAU/mL | 21/230 (9.1%) | 2.04 (0.96–4.33), p=0.11 | 2.39 (1.10–5.22), p=0.06 | 2.55 (1.17–5.58), p=0.04 |
| Quintile 5 (highest 20%): ≥164 BAU/mL (reference) | 11/234 (4.7%) | 1.00 | 1.00 | 1.00 |

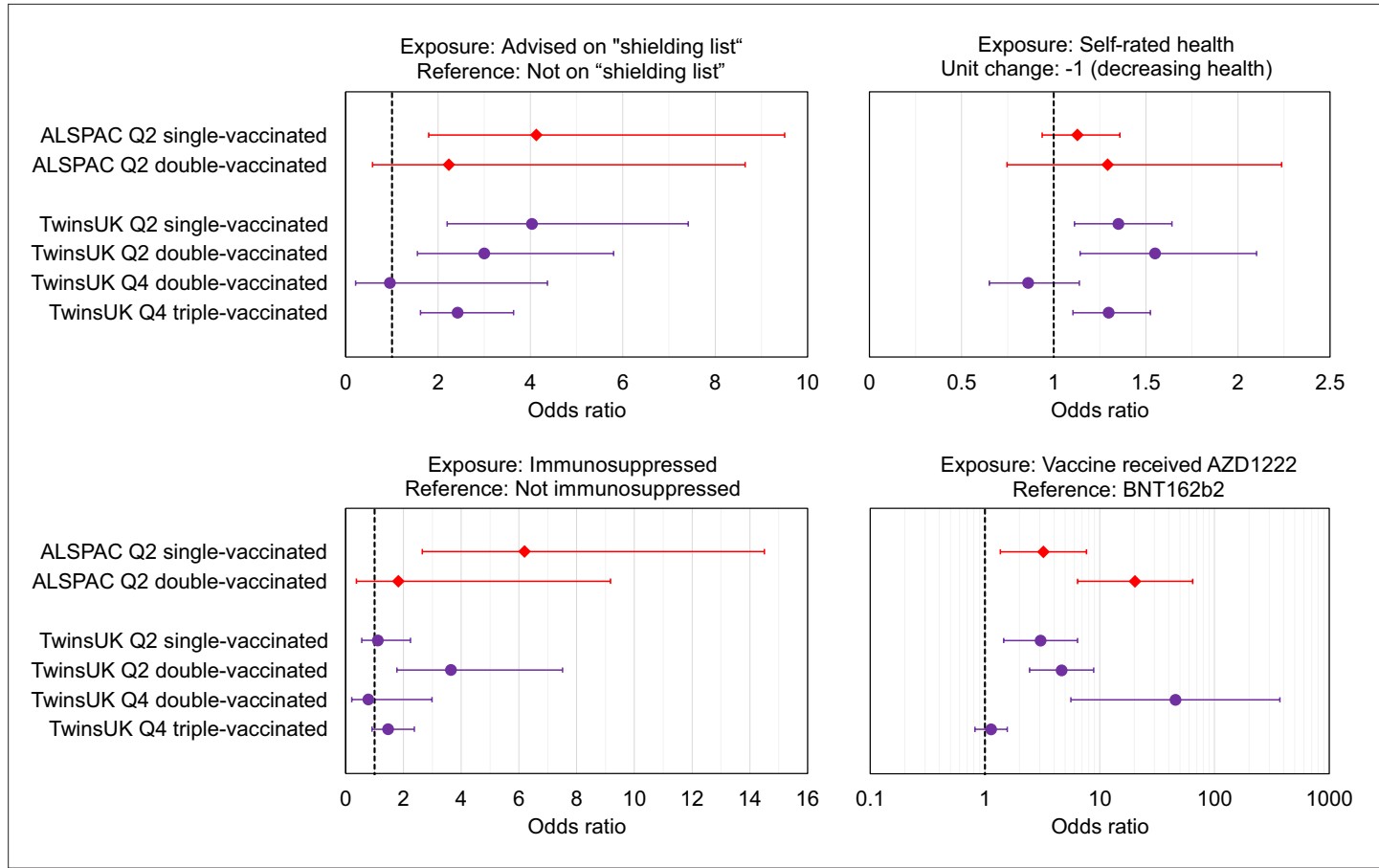

**Figure 3.** Associations with low relative anti-Spike antibody levels within TwinsUK and Avon Longitudinal Study of Parents and Children (ALSPAC). Odds ratios with unadjusted 95% confidence intervals for selected exposure variables, testing associations with low anti-Spike antibody levels, for sub-samples of TwinsUK (purple circles) and ALSPAC (red diamonds) individuals tested in Q2 or Q4, while single-, double-, or triple- vaccinated. Low antibody levels were defined as the lowest 10% within the given sub-sample, except for ALSPAC and TwinsUK Q2 double-vaccinated sub-samples where lowest 8% is used due to assay upper limit. Each point estimate originates from a distinct multivariate logistic regression model, including the exposure variable of interest and adjustment variables of age, sex, name of most recent vaccine received and weeks since most recent vaccination. Note x-axis ranges on sub-plots vary, and vaccine received panel uses a logarithmic x-axis. Odds ratio = 1 is indicated with a dashed black line.

The online version of this article includes the following figure supplement(s) for figure 3:

**Figure supplement 1.** Antibody level differences after third vaccination between related twins and non-related pairs.

---

= 5000 BAU/mL versus 6800 BAU/mL, p=0.0002 for MZ versus DZ), while differences between pairs of non-related individuals were largest (median difference = 7900 BAU/mL, p<0.0001 for MZ versus non-related) (*Figure 3—figure supplement 1*, *Supplementary file 8*).

Generalised linear mixed effects regression models of MZ and DZ twin-pairs were performed with anti-Spike antibody levels after third vaccination as the dependent variable, to further test the persistence of associations between shielding status and antibody levels when shared genetics and early life factors were taken into account. Within MZ twin-pairs discordant for 'Shielded Patient List' status, twins on the 'Shielded Patient List' (within-pair regression coefficient: –3700 BAU/mL, [95% CI: –6500, –880 BAU/mL], p=0.01) had lower antibody levels after third vaccination than their co-twin (*Supplementary file 9*). Between-pair associations with antibody levels were also observed for self-rated health, frailty index, and highest educational attainment, but within-pair coefficients were not significant (*Supplementary file 9*).

## Discussion

In this study we used SARS-CoV-2 anti-Nucleocapsid and anti-Spike antibody testing, and questionnaire data collected at multiple timepoints during and before the COVID-19 pandemic, to investigate associations with antibody response to vaccination in TwinsUK and ALSPAC longitudinal population-based cohorts.

Firstly, we observed large non-linear increases in antibody levels between first, second, and third vaccination, both at the median and 10th percentile levels where risk of infection is heightened, with a relatively narrowed antibody level distribution after third vaccination producing a more even response across the sampled population. Secondly, individuals with lower levels of anti-Spike antibodies following first vaccination were at higher risk of future SARS-CoV-2 infection at any subsequent time, including after further vaccinations, providing further indication of anti-Spike antibody levels as a correlate of protection. Thirdly, the following groups all had higher odds of having lower antibody levels following vaccination: those on the UK 'Shielded Patient List'; those with lower self-rated health; those who received AZD1222 (Oxford/AstraZeneca) vaccine for first and second vaccination; those sampled at longer time since second vaccination and third vaccination; those prescribed immunosuppressant medication (in TwinsUK) or with self-reported immunosuppression (in ALSPAC). These findings were consistent across multiple rounds of vaccination and/or in both cohorts. Individuals with evidence of SARS-CoV-2 infection prior to sampling were less likely to have lower antibody levels, consistent with previous studies that postulating that the quantity and quality of antibody response were linked to the total number of exposures to SARS-CoV-2 (*Walls et al., 2022*; *Bates et al., 2022*). Finally, in analyses exploiting the twin-pair design of the TwinsUK cohort, we found that genetic factors influenced antibody level variation (considered only after third vaccination), with smaller differences in antibody levels within genetically identical MZ pairs compared with DZ pairs. Twin-pair regression models showed that association between antibody levels and 'Shielded Patient List' status was independent of genetic and other shared factors, after explicit adjustment for key vaccination and infection variables.

Longitudinal antibody testing within TwinsUK at Q4 highlighted the effectiveness of third vaccination at both increasing absolute levels of antibodies and reducing variability in post-vaccination antibody levels evident after earlier doses. Even among sub-groups associated with having the lowest antibody levels and/or higher risk of severe COVID-19, such as shielding, frail, and/or immunosuppressed individuals, over 75% of individuals had levels above 6000 BAU/mL (*Supplementary file 4*), the minimum level estimated to give partial protection against omicron BA.1 variant (*Dimeglio et al., 2022b*). Moreover, although individuals receiving AZD1222 vaccine (versus BNT162b2 [Pfizer BioNTech]) were more likely to have lower antibody levels after first and second vaccination, this disparity was no longer evident after third vaccination, consistent with lower vaccine effectiveness and increased post-vaccination infection after first or second vaccination following AZD1222 versus BNT162b2 (*UKHSA, 2021*; *Katikireddi et al., 2022*; *Stouten et al., 2022*), but only minor differences after third vaccination (*UKHSA, 2021*; *Menni et al., 2022*).

Notably, health-related variables associated with lower antibody levels were more general (self-rated poor health, immunosuppression indicators) and/or collective measures with wide-ranging criteria (e.g., 'Shielded Patient List', very frail, multimorbidity), rather than specific factors such as individual comorbidities (e.g., rheumatoid arthritis). These more general and collective measures may contain more specific risk factors for which we did not have data or sufficient sample size to detect, or could suggest that variation in post-vaccination antibody levels between individuals may originate from a wide range of variables *in combination*. Of the several variables associated with antibody levels, only serology-based evidence of prior SARS-CoV-2 infection was directly associated (here, negatively associated) with subsequent post-vaccination infection between April-May 2021 and November 2021-January 2022 (majority sampled before the January 2022 UK omicron variant peak). We found no consistent associations of lower antibody levels with age or employment status, but a very strong age gradient (lower incidence with older age) and lower likelihood among retired (versus employed) individuals of post-vaccination infection. These results are consistent with risk of infection being a complex combination of SARS-CoV-2 case prevalence, individual immune response to vaccination, and individual level of exposure. Given the relaxation of measures across many countries, groups previously less exposed, for example, due to shielding guidance, may become more at risk.

We also acknowledge limitations of this work. Both TwinsUK and ALSPAC (Generation 0) participants are disproportionately older, female, and more likely of white ethnicity, in comparison to the UK population. Geographically, TwinsUK (based in London) is skewed towards lower deprivation areas in south east England and ALSPAC (based in Bristol) towards south west England. Consequently, the generalisability of our findings to non-white UK and international populations, in addition to our ability to detect associations with smaller effect sizes, is limited. Our analyses are subject to selection biases due to use of multiple and varying data collections that rely on voluntary participation. This may cause collider bias and affect findings as outlined elsewhere (*Griffith et al., 2020*; *Munafò et al., 2018*). For example, indicators of poorer health have been associated with lower response to COVID-19 questionnaires in ALSPAC (*Fernández-Sanlés et al., 2021*), which may bias the observed results. Acknowledging the potential effects of biases, the replication of multiple associations with lower antibody levels across compositionally varied TwinsUK and ALSPAC cohorts and across multiple rounds of vaccination support the robustness of our findings. It is these replicated findings that we chose to discuss primarily.

In conclusion, our results highlight the large boost across the antibody level distribution produced by third vaccination, and suggest that measurement of anti-Spike antibodies after first SARS-CoV-2 vaccination may have potential use as an early indicator to identify individuals at higher risk of a future SARS-CoV-2 infection, particularly in the many countries where vaccination roll-out is at an earlier stage. Individuals who previously met UK 'Shielded Patient List' criteria had consistently lower antibody responses to vaccination than other participants, highlighting the importance of continuing to inform such individuals of their personal risk of SARS-CoV-2 infection, despite the UK government decision to end shielding guidance in April 2021 (*GOV.UK, 2021*). This result should inform prioritisation of vaccination towards these individuals in any future immunisation campaigns.

## Data availability

Data from all analyses presented in figures and tables herein are tabulated and available as a supplementary spreadsheet file. Original antibody test data are available within the UK Longitudinal Linkage Collaboration upon application (see https://ukllc.ac.uk/apply/). UK LLC houses COVID-19 related datasets from over 20 UK longitudinal population studies (see https://ukllc.ac.uk/datasets/). Original TwinsUK data are available to researchers on application. Access to original TwinsUK data is managed by the TwinsUK Resource Executive Committee (see https://twinsuk.ac.uk/resources-for-researchers/access-our-data/) and access to original ALSPAC data via an online proposal system (see http://www.bristol.ac.uk/media-library/sites/alspac/documents/researchers/data-access/ALSPAC_Access_Policy.pdf). This is to ensure privacy and protect against misuse. ALSPAC study data were collected and managed using REDCap electronic data capture tools hosted at the University of Bristol. REDCap (Research Electronic Data Capture) is a secure, web-based software platform designed to support data capture for research studies (*Harris et al., 2009*). The study website contains details of all the data that is available through a fully searchable data dictionary and variable search tool on the study website (*University of Bristol, 2022*). Analysis code is available via GitHub at: https://github.com/nathan-cheetham/NCS_SARSCOV2_Antibody_Study, (*Cheetham, 2023* copy archived at swh:1:rev:7103349369b7a4f00d6cf73f780283ba86fe8570).

## Acknowledgements

We are grateful to participants from all studies for participating in testing and completing questionnaires during the COVID-19 pandemic and thank colleagues (TwinsUK: Gulsah Akdag, Andy Anastasiou, Julia Brown, Rachel Horsfall, Genevieve Lachance, Ayrun Nessa, Timothy Spector, Dovile Vaitkute, Sam Wadge, Sivasubramaniam Rajan Wignarajah, Darioush Yarand. ALSPAC: Amy-Louise Gordon, Alix Groom, Nicholas Wells, Hannah Woodward) for coordinating and undertaking data collections and visits during the COVID-19 pandemic.

We are extremely grateful to all the families who took part in this study, the midwives for their help in recruiting them, and the whole ALSPAC team, which includes interviewers, computer and laboratory technicians, clerical workers, research scientists, volunteers, managers, receptionists, and nurses.

NJT is a Wellcome Trust Investigator ([202802/Z/16/Z]), is the PI of the Avon Longitudinal Study of Parents and Children (MRC and WT [217065/Z/19/Z]), is supported by the University of Bristol

NIHR Biomedical Research Centre ([BRC-1215–2001]), the MRC Integrative Epidemiology Unit ([MC_UU_00011/1]), and works within the CRUK Integrative Cancer Epidemiology Programme ([C18281/A29019]). MK is supported by the Medical Research Council ([MR/W021315/1]). AK is supported by Characterisation, determinants, mechanisms, and consequences of the long-term effects of COVID-19: providing the evidence base for health care services (CONVALESCENCE) funded by NIHR ([COV-LT-0009]). SVK acknowledges funding from an NRS Senior Clinical Fellowship ([SCAF/15/02]), the Medical Research Council ([MC_UU_00022/2]), and the Scottish Government Chief Scientist Office ([SPHSU17]). OKLH is supported by the Medical Research Council ([MC_UU_12017/11] and [MC_UU_00022/3]) and the Scottish Government Chief Scientist Office ([SPHSU17]). This publication is the work of the authors and NJC, NJT, and CJS serve as guarantors for the contents of this paper.

## Additional information

### Competing interests

Milla Kibble: received payment for attending the Health and Safety Executive (HSE) symposium. The author has no other competing interests to declare. Charis Bridger Staatz: received an ESRC and NIHR funded Grant, and MRC Funded Studentship. The author has no other competing interests to declare. Srinivasa Vittal Katikireddi: participates on the Scottish Government Expert Reference Group on Ethnicity and COVID-19, and UK Scientific Advisory Group on Emergencies (SAGE) subgroup on Ethnicity. The author has no other competing interests to declare. Maria Paz Garcia: is a member of the King's College London Health Faculties Research Ethics Subcommittee (Purple), and a Chair of the TwinsUK Volunteer Advisory Panel. The author has no other competing interests to declare. Nishi Chaturvedi: received payment for clinical trials of a diabetes drug from AstraZeneca. Nishi Chaturvedi is Chair of British Heart Foundation Fellowships Committee, a member of Diabetes UK research committee and a member of NWO Gravitational Awards Committee. The author has no other competing interests to declare. Claire J Steves: received payment for consultancy work for Zoe Ltd. The author has no other competing interests to declare. The other authors declare that no competing interests exist.

### Funding

| Funder | Grant reference number | Author |
|---|---|---|
| NIHR | COV-LT-0009 | Nathan J Cheetham |
| NIHR Bristol Biomedical Research Centre | BRC-1215-2001 | Nicholas J Timpson |
| MRC Integrative Epidemiology Unit | MC_UU_00011/1 | Nicholas J Timpson |
| Medical Research Council | MR/W021315/1 | Milla Kibble |
| NRS | SCAF/15/02 | Srinivasa Vittal Katikireddi |
| Medical Research Council | MC_UU_00022/2 | Srinivasa Vittal Katikireddi |
| Scottish Government Chief Scientist Office | SPHSU17 | Olivia KL Hamilton |
| Medical Research Council | MC_UU_12017/11 | Olivia KL Hamilton |
| Medical Research Council | MC_UU_00022/3 | Olivia KL Hamilton |

The funders had no role in study design, data collection and interpretation, or the decision to submit the work for publication.

### Author contributions

Nathan J Cheetham, Conceptualization, Data curation, Formal analysis, Funding acquisition, Investigation, Visualization, Methodology, Writing – original draft, Project administration, Writing – review and editing; Milla Kibble, Conceptualization, Data curation, Formal analysis, Investigation, Visualization, Methodology, Writing – original draft, Project administration, Writing – review and editing; Andrew

Wong, Conceptualization, Methodology, Writing – original draft, Project administration, Writing – review and editing; Richard J Silverwood, Methodology, Writing – original draft, Writing – review and editing; Anika Knuppel, Dylan M Williams, Olivia KL Hamilton, Paul H Lee, Charis Bridger Staatz, Giorgio Di Gessa, Jingmin Zhu, Srinivasa Vittal Katikireddi, George B Ploubidis, Ruth E Mitchell, Kate Northstone, Methodology, Writing – review and editing; Ellen J Thompson, Ruth CE Bowyer, Xinyuan Zhang, Golboo Abbasian, Daniel Major-Smith, Thomas Breeze, Michael Crawford, Data curation, Writing – review and editing; Maria Paz Garcia, Deborah Hart, Funding acquisition, Project administration, Writing – review and editing; Jeffrey Seow, Carl Graham, Neophytos Kouphou, Sam Acors, Michael H Malim, Katie Doores, Investigation, Writing – review and editing; Sarah Matthews, Alex SF Kwong, Writing – review and editing; Lynn Molloy, Project administration, Writing – review and editing; Nishi Chaturvedi, Funding acquisition, Methodology, Project administration, Writing – review and editing; Emma L Duncan, Funding acquisition, Writing – original draft, Writing – review and editing; Nicholas J Timpson, Claire J Steves, Conceptualization, Funding acquisition, Methodology, Writing – original draft, Project administration, Writing – review and editing

### Author ORCIDs
Nathan J Cheetham ![ORCID] http://orcid.org/0000-0002-2259-1556
Milla Kibble ![ORCID] http://orcid.org/0000-0003-1130-4010
Olivia KL Hamilton ![ORCID] http://orcid.org/0000-0002-5874-0058
Paul H Lee ![ORCID] http://orcid.org/0000-0002-5729-6450
Charis Bridger Staatz ![ORCID] http://orcid.org/0000-0002-2872-6968
Giorgio Di Gessa ![ORCID] http://orcid.org/0000-0001-6154-1845
Jingmin Zhu ![ORCID] http://orcid.org/0000-0001-8325-7589
Sam Acors ![ORCID] http://orcid.org/0000-0001-6428-7707
Michael H Malim ![ORCID] http://orcid.org/0000-0002-7699-2064
Daniel Major-Smith ![ORCID] http://orcid.org/0000-0001-6467-2023
Emma L Duncan ![ORCID] http://orcid.org/0000-0002-8143-4403
Claire J Steves ![ORCID] http://orcid.org/0000-0002-4910-0489

### Ethics
The ethics statements for each of the longitudinal studies involved in this study are outlined below. TwinsUK: All waves of TwinsUK have received ethical approval associated with TwinsUK Biobank (19/NW/0187), TwinsUK (EC04/015) or Healthy Ageing Twin Study (H.A.T.S) (07/H0802/84) studies from HRA/NHS Research Ethics Committees. The TwinsUK Resource Executive Committee (TREC) oversees management, data sharing and collaborations involving the TwinsUK registry (for further details see https://twinsuk.ac.uk/resources-forresearchers/access-our-data/), in consultation with the TwinsUK Volunteer Advisory Panel (VAP) where needed. ALSPAC: Ethical approval for the study was obtained from the ALSPAC Ethics and Law Committee and the Local Research Ethics Committees. Informed consent for the use of data collected via questionnaires and clinics was obtained from participants following the recommendations of the ALSPAC Ethics and Law Committee at the time. Consent for biological samples has been collected in accordance with the Human Tissue Act (2004). USoc: The University of Essex Ethics Committee has approved all data collection for the Understanding Society main study and COVID-19 web and telephone surveys (ETH1920-1271). The March 2021 web survey was reviewed and ethics approval granted by the NHS Health Research Authority, London - City & East Research Ethics Committee (reference 21/HRA/0644). No additional ethical approval was necessary for this secondary data analysis. 1958 NCDS, 1970 BCS70, Next Steps, MCS: The most recent sweeps of 1958 NCDS, 1970 BCS, Next Steps and MCS have all been granted ethical approval by the National Health Service (NHS) Research Ethics Committee and all participants have given informed consent. ELSA: Waves 1-9 of ELSA were approved by the London Multicentre Research Ethics Committee (approval number MREC/01/2/91),and the COVID-19 substudy was approved by the University College London Research Ethics Committee (0017/003). All participants provided informed consent. 1946 NSHD: Ethical approval for the study was obtained from the NHS Research Ethics Committee (19/LO/1774). All participants provided informed consent. SABRE: Ethical approval for the study was obtained from the NHS Research Ethics Committee (19/LO/1774). All participants provided informed consent. EXCEED: The original EXCEED study was approved by the Leicester Central Research Ethics Committee (Ref: 13/EM/0226). Substantial amendments have been approved by the same Research Ethics Committee for the collection of

new data relating to the COVID-19 pandemic, including the COVID-19 questionnaires and antibody testing.

### Decision letter and Author response
Decision letter https://doi.org/10.7554/eLife.80428.sa1
Author response https://doi.org/10.7554/eLife.80428.sa2

## Additional files

### Supplementary files
• Supplementary file 1. Information on origin of variables used in TwinsUK and Avon Longitudinal Study of Parents and Children (ALSPAC) analysis.

• Supplementary file 2. Anti-Spike antibody level values and characteristics for individuals from TwinsUK sampled in Q2 and Q4 antibody collections. Individuals are stratified by vaccination status at time of sampling. Data shown for individuals sampled at least 4 (2) weeks after first (second or third) vaccination. The antibody level assay range is 0.4–250 BAU/mL for Q2 results and 0.4–25000 BAU/mL for Q4 results, with a positive threshold of 0.8 BAU/mL. Categories with fewer than five individuals are suppressed.

• Supplementary file 3. Severe acute respiratory syndrome coronavirus 2 (SARS-CoV-2) infection prevalence rates, split by selected socio-demographic variables, for TwinsUK Q4 antibody testing participants. p-Values are generated from chi-square test of independence on cross tabulation of counts for the socio-demographic variable of interest and all categories (including those not presented) of the SARS-CoV-2 infection variable.

• Supplementary file 4. Anti-Spike antibody levels and weeks since most recent vaccination within TwinsUK and Avon Longitudinal Study of Parents and Children (ALSPAC) individuals, stratified by vaccination status at Q2 and Q4 antibody testing, split by various variables. The antibody level assay range is 0.4–250 BAU/mL for Q2 results and 0.4–25000 BAU/mL for Q4 results, with a positive threshold of 0.8 BAU/mL.

• Supplementary file 5. Descriptive statistics relating to post-vaccination infections within TwinsUK, within groups of individuals with varying vaccination status at Q2 and Q4 testing.

• Supplementary file 6. Logistic regression model results, testing for association between post-vaccination infection and socio-demographic, severe acute respiratory syndrome coronavirus 2 (SARS-CoV-2) vaccination, and SARS-CoV-2 infection variables for TwinsUK individuals who participated in antibody testing at Q2 and one or both of Q4 antibody testing and Q4 questionnaire, who reported one or more vaccination reported by Q4. Results present odds ratios, unadjusted 95% confidence intervals, and p-values adjusted for multiple testing. Results based on fewer than three individuals having post-vaccination infection are suppressed. Variables with adjusted p-values <0.05 are highlighted in bold.

• Supplementary file 7. Logistic regression model results, testing for association with low anti-Spike antibody levels after first, second, and third severe acute respiratory syndrome coronavirus 2 (SARS-CoV-2) vaccination within TwinsUK and Avon Longitudinal Study of Parents and Children (ALSPAC) at Q2 or Q4 testing. Results present odds ratios, unadjusted 95% confidence intervals, and p-values adjusted for multiple testing. Results based on fewer than three individuals being in the low antibody level group are suppressed. Sets of adjustment variables included in addition to the exposure variable in each model were age, sex, most recent vaccine received and weeks since most recent vaccination, aside from cases where the effect of adjustment variables were themselves tested. In these cases, all other adjustment variables within the given set were included in addition to the adjustment variable being tested. Variables with adjusted p-values <0.05 are highlighted in bold.

• Supplementary file 8. Descriptive statistics of differences in anti-Spike antibody levels between pairs after third severe acute respiratory syndrome coronavirus 2 (SARS-CoV-2) vaccination within TwinsUK. Pair differences are calculated between all complete pairs of monozygotic (MZ) twins and/or dizygotic (DZ) twins, and all combinations of non-related pairs.

• Supplementary file 9. Results of generalised linear mixed effects models testing association with anti-Spike antibody levels after third severe acute respiratory syndrome coronavirus 2 (SARS-CoV-2) vaccination within and between twin-pairs within TwinsUK. Coefficients with unadjusted 95% confidence intervals and unadjusted p-values are presented. Family structure is included as a

random effect, allowing intercepts to vary between twin-pairs. Models are adjusted for age, sex, weeks since third vaccination, third vaccine received, and serology-based infection status. Variables with (two-sided) p-values <0.05 are highlighted in bold.

- MDAR checklist

### Data availability

Data from all analyses presented in figures and tables herein are tabulated and available as a supplementary spreadsheet file. Original antibody test data are available within the UK Longitudinal Linkage Collaboration upon application (see https://ukllc.ac.uk/apply/). UK LLC houses COVID-19 related datasets from over 20 UK longitudinal population studies (see https://ukllc.ac.uk/datasets/). Original TwinsUK data are available to researchers on application. Access to original TwinsUK data is managed by the TwinsUK Resource Executive Committee (see https://twinsuk.ac.uk/resources-for-researchers/access-our-data/) and access to original ALSPAC data via an online proposal system (see http://www.bristol.ac.uk/media-library/sites/alspac/documents/researchers/data-access/ALSPAC_Access_Policy.pdf). This is to ensure privacy and protect against misuse. ALSPAC study data were collected and managed using REDCap electronic data capture tools hosted at the University of Bristol. REDCap (Research Electronic Data Capture) is a secure, web-based software platform designed to support data capture for research studies (doi:https://doi.org/10.1016/J.JBI.2008.08.010). The study website contains details of all the data that is available through a fully searchable data dictionary and variable search tool on the study website (http://www.bristol.ac.uk/alspac/researchers/our-data/). Analysis code is openly available via GitHub at: https://github.com/nathan-cheetham/NCS_SARSCOV2_Antibody_Study (copy archived at swh:1:rev:7103349369b7a4f00d6cf73f780283ba86fe8570).

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
