## [Editor Report]

This important study provides convincing evidence that a third SARS-CoV-2 vaccination elicits substantial increases in spike antibody responses, decreasing variability in titers observed after first and second doses, and also decreasing differences between groups at low and high-risk of low antibody responses. High antibody titers are subsequently associated with a lower incidence of infection. This paper is strong methodologically and will be of interest to clinicians and public health officials.

---

## [Decision Letter]

**Decision letter after peer review:**

Thank you for submitting your article "Antibody levels following vaccination against SARS-CoV-2: associations with post-vaccination infection and risk factors in two UK longitudinal studies" for consideration by *eLife*. Your article has been reviewed by 2 peer reviewers, one of whom is a member of our Board of Reviewing Editors, and the evaluation has been overseen by Miles Davenport as the Senior Editor. The reviewers have opted to remain anonymous.

Essential revisions:

1) Please shorten the paper and focus on its core scientific conclusions.

2) Please only include figures which most succinctly justify these conclusions which include multivariate analyses.

3) Please offer greater detail regarding the analyses as suggested by Reviewer 1.

4) Discuss the relevance of the paper to the current stage of the pandemic with novel VOCs.

*Reviewer #1 (Recommendations for the authors):*

1) The messaging is too diffuse. The Results section of the abstract provides a nice template for the focus of the paper's conclusions, but other sections (the first few paragraphs of the discussion for example) highlight other conclusions. I suggest focusing on how the 3rd vaccine minimized discrepancies between protection among subgroups and vaccine (AZ vs Pfzer) as the singular finding of the paper.

2) The paper needs to be rewritten with an earlier and more consistent focus on whether its findings have any relevance for the current pandemic situation in which omicron and its downstream VOCs predominate. In particular, are Q4 results related to omicron infections exclusively? Does the ALSPAC cohort have any relevance to current conditions? I sympathize that it is challenging to maintain scientific relevance with these types of analyses given the speed and frequency of novel VOC takeover. Nevertheless, it is no longer of high public health relevance to understand whether a single vaccine dose protects against the δ variant. As written, the paper provides no easy roadmap for public health officials to interpret and use its results for vaccine scheduling and prioritization. Moreover, the authors' claim that the paper's findings are relevant as mathematical model inputs is only true if they are relevant for currently dominant VOCs.

3) All conclusions that focus on univariate analyses should be downplayed and removed from the main body of the paper (Figure 3 and 4). I would suggest redoing these figures (with infection as the outcome) in a fashion like Figure 5 with a multivariate analysis. This would provide more compelling evidence that antibody levels protect against infection regardless of other potential exposure variables (age, sex, time since vaccine, etc….). The data for this more informative figure is partially listed in part in the 2nd paragraph on page 12 (OR=2.9, p=0.02) but should be in a figure.

4) Figure 3 would also be much more relevant and interesting if it included data after the 2nd shot and 3rd shot (either in aggregate or as separate panels). I might also consider a separate analysis with antibody levels (rather than quintiles) based on cut-off levels thought to be protective against omicron.

5) For Figure 5, I would choose one cutoff (10%) as the outcome variable to make this very key figure easier to read and less busy. Also, what is the baseline comparator for OR=1? It is not clear from the legend.

6) More detail is needed in the discussion on how much of a shortcoming the predominant white ethnicity is in terms of generalizing results to the UK or to other countries.

7) The conclusion that the variability of antibody levels after the 3rd vaccine decreased relative to the first vaccine requires statistical testing and verification rather than an eyeball assessment of the graph. I am not sure how this can be done based on the lack of a full quantitative range of anti-spike levels at Q2 (see point 9). In this case, I do not think the conclusions are currently justified by the current level of analysis.

8) Overall, the variability of antibody levels is not adequately demonstrated in the figures. First, the x-axes in Figure 1, and y-axes in Figures2 should be of equivalent scale (0-25000) to allow comparison across panels. Second, these axes should be log converted if this normalizes the data. Third, rather than just showing the 10% percentile, include standard boxplots which naturally show the IQR and values within the IQR so that the median, variability, and skew of the data can be visualized. Finally, please reveal what percentage of samples in each Q2 group fall at or above 250 to see if it is reasonable to not test for levels in these groups based on the "assay threshold (see point 9)".

9) Overall, the reasoning for using the assay threshold requires greater quantitative heft and justification.

10) Is there any data between 28 and 34 weeks after the 3rd vaccine to match after the second? This would solidify conclusions about the importance of the 3rd shot.

11) For context, references citing the percentage of the studied population with 1, 2, 3, or 4 vaccines should be included.

12) The "shielded patient list" is a valid exposure variable as it is convenient and apparently well known. However, it contains multiple more biologically relevant exposure variables that likely could better predict risk. This should be acknowledged and outlined in more detail in the discussion.

Other issues

1) 3rd intro paragraph: the list of risk factors should be expanded or contracted as it currently includes broad categories of diseases and very specific diseases (vasculitis).

2) Last paragraph intro: the word "origins" is too strong to explain the analysis of post-vaccine antibody level variability as it implies causality. A better phrase would be "variables associated with".

3) The comparison of median antibody levels among unvaccinated versus vaccinated persons on page 12 is not terribly interesting and the differences are slight (40 vs 57 BAU/ml median) despite a p-value that seems surprisingly very low (0.001) and should be double-checked.

4) Anti-spike antibody is referred to as a proxy for the immune response. This is not correct. The immune response is a broad term as immunity protects against infection but also modifies infection if there is no protection. Antibodies are thought to play somewhat more of a role in the former than the latter. It is better to say that antibodies have been identified as a correlate of protection and cite the appropriate articles.

*Reviewer #2 (Recommendations for the authors):*

(1) The manuscript is a bit too long, with many supplementary materials. I have not checked those SI materials in detail.

(2) Not sure if it is useful to stratify results by the vaccination type. For example, older individuals may be more likely to receive the AstraZeneca vaccine, whereas younger individuals may mainly receive mRNA vaccines. Could you examine if the antibody levels were independent of the type of vaccines?

(3) It is unclear if there were location-specific differences in the results. I understand that the authors analyzed rural-urban differences. But different cities may have some differential patterns.

(4) In the multivariable regression analysis, the authors examined many potential predictor variables. It might be useful to perform a collinearity test to first identify the most useful variables. For example, hierarchical clustering analysis using Ward's method [1] might be helpful.

Ref: [1] Ward's hierarchical agglomerative clustering method: which algorithms implement Ward's criterion? J Classif, 31 (2014), pp. 274-295

---

## [Author Response]

Essential revisions:1) Please shorten the paper and focus on its core scientific conclusions.

Where at all possible we have removed text which is not immediately aligned to the main findings, figures and sections from both the main text (removing 1000 words) and supplementary information (removed 20 pages) to shorten the paper. We have revised introduction and Discussion sections in particular to focus more on the core conclusions. We hope the revised manuscript is therefore improved.

We have made substantive adjustments to the manuscript in efforts to address this point (which was also brought up by reviewers). We summarise key changes below (page numbers refer to when track changes set to ‘All Markup’):

– Introduction (page 6-7) has been revised to reduce length and update sections covering anti-Spike antibody levels as a correlate of protection against future infection and emerging variants, adding references to more recent work, as requested in Editor point (4) and Reviewer 1 point (2).

– Results:

– In “Cohort characteristics” subsection (page 8), paragraphs covering secondary results on infection prevalence across different subgroups and in other longitudinal cohorts have been condensed.

– In “Antibody levels after first, second, and third vaccination” section (page 8), text was revised based on Reviewer 1 point (7) – see below for full response to this point.

– In “Factors associated with post-vaccination infection in TwinsUK” section:

– Text relating to secondary results of sociodemographic associations with post-vaccination infection (page 10) has been condensed.

– Text referring to results testing association between post-vaccination infection and antibody levels (within double-vaccinated participants page 11) has been removed.

– In “Factors associated with lower antibody levels within TwinsUK and ALSPAC” section:

– Text relating to models testing association with the lowest 5% of antibody levels has been removed (page 11-12), to simplify message as suggested by reviewer 1 (point 5).

– Text added to justify choice of threshold antibody level outcome in regression models (page 10), in response to reviewer 1 (point 9) – see below for full response to this point.

– Discussion:

– Paragraph starting “Longitudinal antibody testing…” (page 15) has been moved to earlier in the discussion, as the text describes primary conclusions. This paragraph has been condensed, but also revised to refer to antibody levels in the context of correlates of protection for the omicron variant, as suggested by Editor (point 4) and Reviewer 1 (point 2).

– Paragraph starting “The cross-sectional nature of antibody testing…” (page 16) discussing certain strengths of the study has been removed to condense the text, as it does not add to the primary conclusions of the paper.

– We have removed some text on limitations in the paragraph starting “We also acknowledge limitations of this work” (page 16-17), some of which is now covered earlier in the discussion in revised text, and other text which is not relevant to the primary conclusions of the paper.

– Supplementary information:

– “Shielded patient list criteria” section (page 12-13) has been removed and replaced with a reference in the main text (reference 30) to a website which provides equivalent information

– “All cohort antibody testing summary” section (page 16-17) has been removed, as associated text in main text has been removed.

– In “Factors associated with post-vaccination infection” section (page 29), Table S 7 in our initial submission has been removed, as data included is covered sufficiently by main text Table 3.

– In “Factors associated with post-vaccination infection” section (page 34), Figure S 6 in our initial submission is removed as data is no longer referred to in main text.

– In “Factors associated with low antibody levels: logistic regression results” section (page 35-39), Table S 9 in our initial submission (Supplementary file 7 in revised submission) results columns for models testing association with the lowest 5% of antibody levels are removed, to simplify message as suggested by reviewer 1 (point 5).

– Sections “Factors associated with low antibody levels: TwinsUK figures” (page 47-53) and “Factors associated with low antibody levels: Age, sex and BMI (combined ALSPAC and TwinsUK)” (page 54) have been removed, as the figures duplicate results presented in Table S 9 in our initial submission (Supplementary file 7 in revised submission) and are removed to condense the manuscript.

2) Please only include figures which most succinctly justify these conclusions which include multivariate analyses.

We have made the following changes to figures:

a. Figure 1 (page 40) and Figure 2 (page 43) has been revised based on feedback given by Reviewer 1, making the following changes: (1) box plot layers have been added to visualise variability; (2) antibody level axis range is changed to logarithmic scale and is the same for all subplots; (3) the percentage of samples with values exceeding the assay limit is displayed in figure 1; (4) Lines illustrating the assay upper limit are added for clarity.

b. Former Figures 3 and 4 (page 45-46) which presented results of univariate analysis have been removed as part of condensing the paper, as we do not believe they add to key conclusions.

c. Figure 5 (now Figure 3, page 47) has been modified based on reviewer 1 (point 5) suggestion, removing results based on the lower 5% cut-off. Results and discussion have been changed accordingly. The titles of the sub-plots have been amended to show more clearly the reference used.

3) Please offer greater detail regarding the analyses as suggested by Reviewer 1.

We have revised the manuscript based on reviewer 1 feedback and responded individually to each point given by reviewer 1 below, giving specific references to changes in the manuscript.

4) Discuss the relevance of the paper to the current stage of the pandemic with novel VOCs.

We have made revisions to introduction and Discussion sections and give the specific references to changes in the manuscript in response to the relevant comments from Reviewer 1 below.

Reviewer #1 (Recommendations for the authors):1) The messaging is too diffuse. The Results section of the abstract provides a nice template for the focus of the paper's conclusions, but other sections (the first few paragraphs of the discussion for example) highlight other conclusions. I suggest focusing on how the 3rd vaccine minimized discrepancies between protection among subgroups and vaccine (AZ vs Pfzer) as the singular finding of the paper.

Many thanks for this comment. We have made extensive changes to the manuscript which are highlighted above in our response to Editor (point 1) where we list changes made to address this point.

2) The paper needs to be rewritten with an earlier and more consistent focus on whether its findings have any relevance for the current pandemic situation in which omicron and its downstream VOCs predominate. In particular, are Q4 results related to omicron infections exclusively? Does the ALSPAC cohort have any relevance to current conditions? I sympathize that it is challenging to maintain scientific relevance with these types of analyses given the speed and frequency of novel VOC takeover. Nevertheless, it is no longer of high public health relevance to understand whether a single vaccine dose protects against the δ variant. As written, the paper provides no easy roadmap for public health officials to interpret and use its results for vaccine scheduling and prioritization. Moreover, the authors' claim that the paper's findings are relevant as mathematical model inputs is only true if they are relevant for currently dominant VOCs.

With respect to “focus on whether its findings have any relevance for the current pandemic situation in which omicron and its downstream VOCs predominate” and “it is no longer of high public health relevance to understand whether a single vaccine dose protects against the δ variant”, the primary focus of our results is on identifying groups more likely to have relatively lower levels of antibodies in response to vaccination. As we reference in the revised introduction (page 6, references 19-26), previous studies have repeatedly shown that antibody levels are inversely correlated with risk of future infection for all variants of concern throughout the COVID-19 pandemic. While we focus on relative antibody levels for these reasons, we have updated the introduction to include recent references which estimate protective threshold levels of anti-Spike antibodies, including for the omicron variant (reference 25, doi:10.3390/vaccines10091548). We then comment on our results in the context of these estimated thresholds for omicron, to make our results more relevant to the current dominant variant.

Similarly, with respect to “roadmap for public health officials to interpret and use its results”, we believe that the identification of certain groups as having consistently lower antibody levels across multiple rounds of vaccination is useful information for vaccine prioritisation and is independent of the dominant variant at a given time (given vaccine prioritisation itself is based on relative risk between groups). Knowledge of which groups of the population are more likely to have lower levels after vaccination is useful important indicator in public health of who is more likely to be at risk of infection, and so who could be prioritised in future vaccination rounds – a measure that could be a useful adjunct to antigen testing which retains use in the assessment of case status, but is less able to project the maintenance of circulating antibody response (albeit only one part of immunological response).

With respect to “Does the ALSPAC cohort have any relevance to current conditions?”, the primary purpose of the ALSPAC cohort results is to serve as an independent and parallel analysis of key findings within the primary cohort; namely that certain groups are more likely to have the lowest levels of antibodies following vaccination, therefore we believe it is important to keep these results in the manuscript.

The paragraph of the discussion where we stated that our serological results could be useful for mathematical modelling (page 16) has been removed as part of condensing the section as we decided it was not essential.

3) All conclusions that focus on univariate analyses should be downplayed and removed from the main body of the paper (Figure 3 and 4). I would suggest redoing these figures (with infection as the outcome) in a fashion like Figure 5 with a multivariate analysis. This would provide more compelling evidence that antibody levels protect against infection regardless of other potential exposure variables (age, sex, time since vaccine, etc….). The data for this more informative figure is partially listed in part in the 2nd paragraph on page 12 (OR=2.9, p=0.02) but should be in a figure.

Thank you for this suggestion. We see value in this shift of focus and have adjusted the paper to concentrate on multivariable model results – though not remove already reported univariable results. With respect to “The data for this more informative figure is partially listed in part in the 2nd paragraph on page 12 (OR=2.9, p=0.02) but should be in a figure”, the results of multivariable analyses in question are given in Table 3. Figures 3 and 4 (page 45-46) which presented results of univariate analysis have been removed as part of condensing the paper, as we do not believe they add to main conclusions.

4) Figure 3 would also be much more relevant and interesting if it included data after the 2nd shot and 3rd shot (either in aggregate or as separate panels). I might also consider a separate analysis with antibody levels (rather than quintiles) based on cut-off levels thought to be protective against omicron.

We agree that the proposed analyses would be interesting and useful. Unfortunately, our data does not allow these analyses to be undertaken. Our initial testing was done at a time when individuals had 1 or 2 vaccinations only. Furthermore, the assay limit of this testing round did not allow analysis of risk of infection after 2^nd^ vaccination, only after 1^st^ as presented.

5) For Figure 5, I would choose one cutoff (10%) as the outcome variable to make this very key figure easier to read and less busy. Also, what is the baseline comparator for OR=1? It is not clear from the legend.

Figure 5 has been changed as described in Editor revisions section (point 2) above.

6) More detail is needed in the discussion on how much of a shortcoming the predominant white ethnicity is in terms of generalizing results to the UK or to other countries.

We have revised relevant text in the discussion to explicitly highlight this as limiting the generalisability of our findings to non-white UK and international populations:

Revised text, Discussion, page 17, “Consequently, the generalisability of our findings to non-white UK and international populations, in addition to our ability to detect associations with smaller effect sizes, is limited.”

7) The conclusion that the variability of antibody levels after the 3rd vaccine decreased relative to the first vaccine requires statistical testing and verification rather than an eyeball assessment of the graph. I am not sure how this can be done based on the lack of a full quantitative range of anti-spike levels at Q2 (see point 9). In this case, I do not think the conclusions are currently justified by the current level of analysis.

With respect to “The conclusion that the variability of antibody levels after the 3rd vaccine decreased relative to the first vaccine requires statistical testing and verification rather than an eyeball assessment of the graph”, we have added additional text that compares the ratio of the median to interquartile range as an additional measure of the relative width of the antibody level distributions. We have also added box plots to Figures 1 and 2 to help visualise inter-quartile range as suggested in later reviewer comments.

Revised text, Results, Antibody levels after first, second, and third vaccination subsection, page 9, “The antibody level distribution after third vaccination was relatively narrower compared with earlier vaccination (median:IQR ratios of 0.54, 0.27, and 0.89 among Q2 single-vaccinated, Q4 double-vaccinated, and Q4 triple-vaccinated sub-samples respectively), with smaller scale-factor differences between those with median and lowest levels (median:10^th^ percentile ratios of 5.6, 11.8, and 2.7 among Q2 single-vaccinated, Q4 double-vaccinated, and Q4 triple-vaccinated sub-samples respectively)”

With respect to “I am not sure how this can be done based on the lack of a full quantitative range of anti-spike levels at Q2 (see point 9). In this case, I do not think the conclusions are currently justified by the current level of analysis.”, we do not agree that the assay limit affects our ability to make these calculations and comments – we limit the statement quoted above to comparison of variability between single-, double- and triple-vaccinated sub-samples where the assay limit affects the top 5-22% only, and exclude the Q2 double-vaccinated group from comment because the assay limit does not allow this calculation.

8) Overall, the variability of antibody levels is not adequately demonstrated in the figures. First, the x-axes in Figure 1, and y-axes in Figures2 should be of equivalent scale (0-25000) to allow comparison across panels. Second, these axes should be log converted if this normalizes the data. Third, rather than just showing the 10% percentile, include standard boxplots which naturally show the IQR and values within the IQR so that the median, variability, and skew of the data can be visualized. Finally, please reveal what percentage of samples in each Q2 group fall at or above 250 to see if it is reasonable to not test for levels in these groups based on the "assay threshold (see point 9)".

Figures 1 and 2 have been changed as described in Editor revisions section above.

With respect to “the variability of antibody levels is not adequately demonstrated in the figures”, we have kept the original dot plot layer showing individual values and lines showing the 10^th^ percentile value, as we believe it aids comparison of sample size and allows greater visualisation of values at the lower end of the antibody level distribution (which is a focus of our analysis). Examination of this variation is not the case solely using a box plot as suggested by the reviewer, which focuses on the central 50% of data.

9) Overall, the reasoning for using the assay threshold requires greater quantitative heft and justification.

Text providing additional explanation of the choice to use relative thresholds for antibody levels has been added:

Revised text, Results, Factors associated with lower antibody levels within TwinsUK and ALSPAC section, page 11,

“Lower antibody levels were defined as the lowest 10% within each sub-sample of cohort, testing round and vaccination status (< 250 BAU/mL threshold corresponding to lowest 8% in both TwinsUK and ALSPAC used for Q2 double-vaccinated sub-samples where assay limit did not allow lowest 10% to be identified). Relative thresholds were used rather than absolute values due to the variation in reported thresholds between studies and for different SARS-CoV-2 variants, while the more general principal that antibody levels are inversely correlated with risk of infection has remained consistent throughout the COVID-19 pandemic.”

10) Is there any data between 28 and 34 weeks after the 3rd vaccine to match after the second? This would solidify conclusions about the importance of the 3rd shot.

There is no data available at the requested time points since vaccination, beyond what is shown in the existing figure. This is due to the timing of the sample collection, where that length of time had not passed since 3^rd^ vaccination.

11) For context, references citing the percentage of the studied population with 1, 2, 3, or 4 vaccines should be included.

It is possible to calculate percentages from the sample sizes quoted in Table 1 in the main text and flowcharts Figure S1, S2, S3 in our initial submission (Supplementary files 1, 2, 3 in our revised submission) in the supplementary information. In all efforts to reduce the overall, non-essential, content of the paper we are keen not to add this in at this stage. If there is a persistent editorial need to include this detail, we are of course happy to add this.

12) The "shielded patient list" is a valid exposure variable as it is convenient and apparently well known. However, it contains multiple more biologically relevant exposure variables that likely could better predict risk. This should be acknowledged and outlined in more detail in the discussion.

We have revised an existing section of the discussion which comments on the fact that we saw associations with more general/collective exposures rather than more specific exposures, to include the possibility that the more general/collective exposures we identify as risk factors contain more biologically relevant exposures for which we didn’t have data available to identify directly as suggested by the reviewer.

Revised text, Discussion, page 15, “These more general and collective measures may contain more specific risk factors for which we did not have data for or sufficient sample size to detect, or could suggest that variation in post-vaccination antibody levels between individuals may originate from a wide range of variables *in combination*.”

Other issues1) 3rd intro paragraph: the list of risk factors should be expanded or contracted as it currently includes broad categories of diseases and very specific diseases (vasculitis).

Vasculitis has been removed from list of comorbidities inside brackets have been deleted to avoid inconsistency in specificity of diseases.

Revised text, introduction, page 7, “Lower antibody levels following both first and second vaccinations have been observed in individuals with particular comorbidities (including cancer, renal disease, and hepatic disease [27–29]),…”

2) Last paragraph intro: the word "origins" is too strong to explain the analysis of post-vaccine antibody level variability as it implies causality. A better phrase would be "variables associated with".

The first sentence of the last paragraph of the introduction has been re-written using the reviewer suggestion, replacing ‘origins’ with ‘variables associated with’.

Revised text, introduction, page 7, “Here, we aimed to examine variables associated with variation in post-vaccination antibody levels…”

3) The comparison of median antibody levels among unvaccinated versus vaccinated persons on page 12 is not terribly interesting and the differences are slight (40 vs 57 BAU/ml median) despite a p-value that seems surprisingly very low (0.001) and should be double-checked.

Rather than unvaccinated vs vaccinated groups, the comparison referred to by the reviewer is a comparison of median levels among single-vaccinated individuals at time of testing, comparing individuals who subsequently did not vs did record a SARS-CoV-2 infection in the following 6-9 months. This text has been removed as per earlier reviewer suggestions to condense paper and focus on results of multivariable models.

4) Anti-spike antibody is referred to as a proxy for the immune response. This is not correct. The immune response is a broad term as immunity protects against infection but also modifies infection if there is no protection. Antibodies are thought to play somewhat more of a role in the former than the latter. It is better to say that antibodies have been identified as a correlate of protection and cite the appropriate articles.

The sentence has been revised, replacing previous text “used as a proxy for the humoral immune response” to state that anti-Spike antibody levels have been “identified as a correlate of protection against infection”.

Revised text, Introduction, page 7, “We aimed firstly to assess the relationship between anti-Spike antibody levels (identified as a correlate of protection against infection), measured after first or second vaccination in April-May 2021,…”

Reviewer #2 (Recommendations for the authors):(1) The manuscript is a bit too long, with many supplementary materials. I have not checked those SI materials in detail.

Please refer to the response to Editor (point 1) and also to the comments of Reviewer #1. We have substantially revised the paper and list substantive changes at the top of this response.

(2) Not sure if it is useful to stratify results by the vaccination type. For example, older individuals may be more likely to receive the AstraZeneca vaccine, whereas younger individuals may mainly receive mRNA vaccines. Could you examine if the antibody levels were independent of the type of vaccines?

We agree it is important to note the relationships between personal characteristics and which vaccine was given at different points during the UK vaccination roll-out. We believe that by including age as an adjustment variable in our multivariable model testing the effect of vaccine type on the likelihood of having low antibody levels, we have controlled for the relationship between age and type of vaccine received.

(3) It is unclear if there were location-specific differences in the results. I understand that the authors analyzed rural-urban differences. But different cities may have some differential patterns.

We agree that different regions have different infection patterns, however we did not examine the potential association between geographic region on antibody levels. We anticipate that we would have insufficient numbers of participants from individual city locations to be able to robustly identify differences. We did test association between antibody levels and other sociodemographic variables that are correlated with city demographics and urban vs. rural, for example age, ethnicity and local area deprivation. These results are included in the supplementary information.

(4) In the multivariable regression analysis, the authors examined many potential predictor variables. It might be useful to perform a collinearity test to first identify the most useful variables. For example, hierarchical clustering analysis using Ward's method [1] might be helpful.

We thank reviewer for the suggestion. For this analysis, we decided to test a wide range of exposure variables, to produce a compendium of results that others can refer to when looking for associations with specific socio-demographic and health characteristics e.g. individual comorbidities. So we decided not to try to reduce our exposure variables as part of our analysis protocol.